# Approximation Rate of the Transformer Architecture for Sequence Modeling

**Haotian Jiang** [*]
haotian.jiang@cnrsatcreate.sg

**Qianxiao Li** [*†]
qianxiao@nus.edu.sg

## Abstract

The Transformer architecture is widely applied in sequence modeling applications, yet the theoretical understanding of its working principles remains limited. In this work, we investigate the approximation rate results for the Transformer architectures on general sequence to sequence target relationships. We begin by establishing a representation theorem for the target space and introduce a novel notion of complexity measures to construct approximation spaces. These measures encapsulate both pairwise and pointwise interactions among input tokens. Based on this framework, we derive an explicit Jackson-type approximation rate estimate for the Transformer. This rate sheds light on the underlying structural characteristics of the Transformer, thereby delineating the types of sequential relationships they excel in approximating. Notably, our findings on approximation rates facilitate a concrete comparison between the Transformer and traditional sequence modeling approaches, such as recurrent neural networks.

## 1 Introduction

The Transformer architecture, as introduced by Vaswani et al. [30] has become immensely popular in the field of sequence modeling. Variants such as BERT [8] and GPT [6] have achieved excellent performance, becoming the default choices for natural language processing (NLP) problems. Concurrently, Dosovitskiy et al. [11] successfully applied the Transformer to image classification problems by flattening the image into a sequence of patches. Despite its success across various fields of practical application, many theoretical questions remain unanswered. Among these, we focus on two essential questions in this work: firstly, the approximation rate of the Transformer on sequence modeling; secondly, the comparative advantages and disadvantages of the Transformer with recurrent neural networks (RNNs) on different temporal structures.

The concept of Jackson-type approximation rates is derived from the Jackson Theorem for polynomials [15], and is further elaborated in the work of DeVore [9] for addressing general forward approximation problems. To illustrate this, consider classic polynomial approximation. It involves defining an appropriate approximation space, accompanied by specific complexity measures, precisely the Sobolev space. According to the Jackson theorem, a function with a small Sobolev norm can be efficiently approximated by a polynomial. We aim to establish similar approximation rate results for the Transformer. This identifies the type of targets that the Transformer can efficiently learn. We examine general non-linear sequence-to-sequence target relationships, extending the linear target form explored in previous studies of the approximation results for linear RNNs and linear temporal convolution networks [24, 16]. We introduce a novel notion of complexity, establishing a concrete target space from which approximation rates can be deduced. This enables us to identify the types of sequential relationships that Transformer can approximate efficiently. Based on our theoretical analysis, we identify concrete classes of temporal structures where the Transformer outperforms

---

[*]CNRS@CREATE LTD, 1 Create Way, #08-01 CREATE Tower, Singapore 138602

[†]Department of Mathematics, Institute for Functional Intelligent Materials, National University of Singapore

38th Conference on Neural Information Processing Systems (NeurIPS 2024).

the RNNs and vice versa. Our main contributions are summarized as follows: 1. We develop Jackson-type approximation rate results for single-layer Transformer networks with one attention head. Our analysis reveals that the approximation capacity is governed by a low-rank structure within the pairwise coupling of the target's temporal features. Empirical validation confirms that the findings observed under theoretical settings also hold true in practical applications. 2. We conduct a comparative analysis between the Transformer and RNNs, aiming to identify specific types of temporal structures where one model excels or underperforms compared to the other.

## 2 Related work

We first review the approximation results of Transformer networks. The universal approximation property (UAP) of the Transformer architecture is first proved in Yun et al. [33], which is further extended to Transformers with sparse attention matrices [34]. The above universal results are developed for a deep Transformer structure. In contrast, Kajitsuka & Sato [17] applying a similar technique to prove one layer Transformers can achieve UAP by increasing the width. Additionally, Kratsios et al. [19], Edelman et al. [12], Luo et al. [25] considers the UAP of the Transformer in various different settings. Giannou et al. [13] considers a special setting regarding expressiveness, demonstrating that Transformers can represent any computer program. Beyond the UAP results, there have been developments in specific approximation rate results. Gurevych et al. [14] demonstrated the rate of the misclassification probability by considering the approximation of hierarchical composition functions, which are composed of sparse functions. This rate takes into account both the level of composition and the smoothness of the component functions. Bai et al. [1] and Wang & E [31] explore target relationships with certain special structures. Additionally, Takakura & Suzuki [29] developed approximation rates for a function space consisting of infinite-length sequence-to-sequence functions, which is characterized by the smoothness of the functions. Our approximation rate result steps further by considering temporal structures, shedding light on how the Transformer model handles temporal relationships. Apart from the approximation results, numerous intrinsic properties of the Transformer have been investigated. Dong et al. [10] and Bhojanapalli et al. [4] considers the rank structure of the attention matrices. Levine et al. [20] examine the correlation between the dependency of input variables and the depth of the model. The Transformer is a very flexible architecture, such that a special configuration of parameters can emulate other architectures. For example, Cordonnier et al. [7], Li et al. [21] showed that attention layers under certain assumptions can perform convolution operations. However, not all emulations are valid explanations of the working principles of the Transformer. In this context, definitions of complexity measures and the resulting Jackson-type approximation rate estimates provide more insight into the inner workings of the architecture. This is the focus of the current work.

## 3 Sequence modeling as an approximation problem

We first motivate the theoretical settings of Jackson-type approximation rate results by considering classic polynomial approximation. Then, we formulate the Transformer as an instance of such an approximation problem.

**Motivation of Jackson-type Approximation Rates** Consider two normed vector spaces, $\mathcal{X}$ and $\mathcal{Y}$, designated as the input and output spaces, respectively. We define the target space $\mathcal{C} \subset \mathcal{Y}^{\mathcal{X}}$ as a set of mappings from $\mathcal{X}$ to $\mathcal{Y}$ that we aim to approximate with simpler functions. The hypothesis space is denoted by $\mathcal{H} = \bigcup_m \mathcal{H}^{(m)}$, where $\mathcal{H}^{(m)}$ represents a sequence of hypothesis spaces. Here, $m$ denotes the complexity or the approximation budget of these spaces. Hypothesis space $\mathcal{H}$ encompasses the candidate functions used to approximate targets in $\mathcal{C}$. Let $\alpha$ be a constant, we introduce a complexity measure, denoted as $C^{(\alpha)} : \mathcal{C} \to \mathbb{R}$, based on the structure of $\mathcal{H}$. The complexity measure is used to construct an approximation space $\mathcal{C}^{(\alpha)} := \{H \in \mathcal{C} : C^{(\alpha)}(H) < \infty\}$, which is usually dense in $\mathcal{C}$. Then for any $H \in \mathcal{C}^{(\alpha)}$, the Jackson-type approximation rate is expressed as follows:

$$\inf_{\hat{H} \in \mathcal{H}^{(m)}} \left\| H - \hat{H} \right\| \leq E(C^{(\alpha)}(H), m). \tag{1}$$

Here, the error bound $E(\cdot, m)$ decreases to zero as $m$ approaches infinity, and the rate of decay is usually called the approximation rate. In this context, $C^{(\alpha)}(H)$ quantifies the complexity of a target

$H$ when approximated using $\mathcal{H}$. A smaller value indicates that the target $H$ can be more efficiently approximated with candidates from $\mathcal{H}$. Consequently, the complexity measure discerns the types of targets that can be efficiently approximated within the given hypothesis space. Notably, different hypothesis spaces typically give rise to different complexity measures and approximation spaces. These variations characterize the approximation capabilities of the hypothesis spaces themselves.

Defining an appropriate approximation space $\mathcal{C}^{(\alpha)}$ is essential. Without specific structures, the general target space $\mathcal{C}$ does not provide any rate results. Opting for an approximation space with defined structures allows for the derivation of approximation rates. Moreover, the property that $\mathcal{C}^{(\alpha)}$ is dense in $\mathcal{C}$ ensures that the restriction to $\mathcal{C}^{(\alpha)}$ is not overly limiting, preserving the necessary expressiveness of the space. To illustrate these concepts, we consider polynomial approximation over the interval $[0, 1]$. In this case we set $\mathcal{X} = [0, 1]$ and $\mathcal{Y} = \mathbb{R}$. The target space $\mathcal{C} = C([0, 1])$ is the set of continuous functions defined on $[0, 1]$. The hypothesis space comprises all polynomials, expressed as follows:

$$\mathcal{H} = \bigcup_{m \in \mathbb{N}} \mathcal{H}^{(m)} = \bigcup_{m \in \mathbb{N}} \left\{ \hat{H}(x) = \sum_{k=0}^{m-1} a_k x^k : a_k \in \mathbb{R} \right\}.$$

According to the Jackson theorem for [15], the Sobolev norm serves as an appropriate complexity measure, defined as $C^{(\alpha)}(H) = \max_{r=1\ldots\alpha} \|H^{(r)}\|_\infty$. Let $\mathcal{C}^{(\alpha)}$ be the approximation space containing targets with finite complexity measures. Consequently, for $H \in \mathcal{C}^{(\alpha)}$, we have the following approximate rate:

$$\inf_{\hat{H} \in \mathcal{H}^{(m)}} \|H - \hat{H}\| \leq \frac{c_\alpha}{m^\alpha} C(H). \tag{2}$$

Here, $c_\alpha$ is a constant depending only on $\alpha$. This theorem implies that smooth functions with small Sobolev norms can be efficiently approximated by polynomials. Developing Jackson-type approximation rates for various sequence modeling hypothesis spaces is crucial for understanding their differences. Jackson-type results for RNNs, CNNs, and encoder-decoder hypothesis spaces have been established in Li et al. [24], Jiang et al. [16], Li et al. [23], where complexity measures such as decaying memory and sparsity were found to influence the approximation rates. In Section 4.1, we identify appropriate complexity measures regarding the Transformer and develop corresponding approximation rates. This enables us to discern the essential structures that facilitate efficient approximation using the Transformer. Furthermore, it allows us to understand how and when the Transformer architecture differs from traditional sequence modeling architecture RNNs, which we will discuss in Section 6.

**Formulation of Sequence Modeling as Approximation Problems**  In sequence modeling, we seek to learn relationships between two sequences $\boldsymbol{x}$ and $\boldsymbol{y}$. Mathematically, we consider an input sequence space

$$\mathcal{X} = \left\{ \boldsymbol{x} : x(s) \in [0, 1]^d, \text{ for all } s \in [\tau] \right\}. \tag{3}$$

Here, $[\tau] := \{1, \ldots, \tau\}$ and $\tau$ denotes the maximum length of the input, and we focus on the finite setting, where $\tau < \infty$. Corresponding to each input $\boldsymbol{x} \in \mathcal{X}$ is an output sequence $\boldsymbol{y}$ belonging to

$$\mathcal{Y} = \{ \boldsymbol{y} : y(s) \in \mathbb{R}, \text{ for all } s \in [\tau] \}. \tag{4}$$

We use $\boldsymbol{H} := \{H_t\}_{t=1}^\tau$ to denote the mapping between $\boldsymbol{x}$ and $\boldsymbol{y}$, such that $y(t) = H_t(\boldsymbol{x})$ for each $t \in [\tau]$. Define $C(\mathcal{X}, \mathcal{Y})$ to denote the space of continuous mappings between the input and output space. We may regard each $H_t : [0, 1]^{d \times \tau} \to \mathbb{R}$ as a $\tau$-variable function, where each variable is a vector in $[0, 1]^d$. Next, we define the Transformer hypothesis space.

**The Transformer Hypothesis Space.**  We consider the following Transformer block retaining most of the important components.

$$\hat{H}_t(\boldsymbol{x}) = \hat{F}\left( \sum_{s=1}^\tau \sigma[(W_Q \hat{h}(t))^\top W_K \hat{h}(\cdot)](s) \cdot W_V \hat{h}(s) \right), \tag{5}$$

where $\hat{h} = \hat{f} \circ x$ and $\hat{F} : \mathbb{R}^{m_v} \to \mathbb{R}, \hat{f} : \mathbb{R}^d \to \mathbb{R}^n$ are two feed-forward networks. The parameter matrices have dimension $W_Q, W_K \in \mathbb{R}^{m_h \times n}, W_V \in \mathbb{R}^{m_v \times n}$. The softmax function is denoted as $\sigma$, such that $\sigma[\rho(t, \cdot)](s) = \frac{\exp(\rho(t,s))}{\sum_{s'} \exp(\rho(t,s'))}$. We focus on a simplified architecture: a single-layer

Transformer with one head. In this work, layer normalization and residual connections are not taken into account. While this constitutes a simplified setting intended for theoretical analysis, it's worth noting that the phenomena observed under the theoretical settings also hold true in practical applications, as we have demonstrated in Section 5. The approximation budget of the Transformer depends on several components. We use $\hat{\mathcal{F}}^{(m_{\mathrm{FF}})}$ to denote the class of feed-forward networks used in the Transformer with budget $m_{\mathrm{FF}}$, which is usually determined by the number of neurons and layers. Let $m = (n, m_h, m_v, m_{\mathrm{FF}})$ denote the overall approximation budget. We use $\mathcal{H}^{(m)}$ to denote the Transformer with complexity $m$. Then, we define the Transformer hypothesis space by

$$\mathcal{H} = \bigcup_m \mathcal{H}^{(m)}, \qquad \mathcal{H}^{(m)} = \left\{ \hat{\boldsymbol{H}} : \hat{\boldsymbol{H}} \text{ satisfies Equation (5) with } m \right\}. \tag{6}$$

## 4 Approximation results

Following the motivation of approximation problems for sequence modeling as discussed in Section 3, this section discusses the approximation rate results for the Transformer. Firstly, we introduce the notion of the permutation equivariance property of the Transformer and discuss the role of position encodings in eliminating it. Next, we define the target space and present the corresponding representation theorem. We then establish the complexity measures necessary to form the approximation space. Our main result Theorem 4.2 presents the approximation rate results.

**Permutation Equivariance and Position Encodings**   Our objective is to approximate a target relationship $\boldsymbol{H}$ where each $\boldsymbol{H}$ belongs to $C(\mathcal{X}, \mathcal{Y})$. It is important to note that without specific modifications, the Transformer inherently cannot approximate such targets due to its permutation equivariance properties. This property implies that permuting the temporal indices of the input sequence results in a corresponding permutation of the output sequence. More precisely, if $p$ denotes a bijection on $[\tau]$ representing a permutation of $\tau$ objects, a sequence of functions $\boldsymbol{H}$ is considered permutation equivariant if for all bijections $p$ and inputs $\boldsymbol{x} \in \mathcal{X}$, the condition $\boldsymbol{H}(\boldsymbol{x} \circ p) = \boldsymbol{H}(\boldsymbol{x}) \circ p$ holds. The Transformer $\hat{\boldsymbol{H}}$ within the hypothesis space $\mathcal{H}$ is indeed permutation equivariant (refer to Appendix A for details). Directly applying the Transformer, therefore, yields permutation equivariant hypotheses, which are inadequate for approximating general sequential relationships that lack this symmetry. In practical applications, incorporating position encodings is a widely adopted approach to counteract permutation equivariance [30]. Various methods exist for embedding positional information. The simplest approach is fixed encoding, which involves mapping $x(t)$ to a higher-dimensional space and then offsetting each $x(t)$ by distinct distances. Formally, with $A \in \mathbb{R}^{d' \times d}$ where $d' \geq d$ and a constant or trainable $e(t) \in \mathbb{R}^{d'}$, position encodings can be expressed as $x(t) \mapsto A x(t) + e(t)$. For general purposes, it's sufficient for the encoded input space $\mathcal{X}^{(E)}$ to satisfy the following condition:

$$\mathcal{X}^{(E)} = \left\{ \boldsymbol{x} : x(s) \in \mathcal{I}^{(s)} \subset \mathbb{R}^d, \text{ where } \mathcal{I}^{(i)}, \mathcal{I}^{(j)} \text{ are disjoint, compact } \forall i, j \in [\tau] \right\}. \tag{7}$$

This ensures that for each input $\boldsymbol{x} = (x(1), \ldots, x(\tau))$, all tokens $x(i)$ and $x(j)$ are distinct, meaning no two input sequences are temporal permutations of each other. Define the set $\mathcal{I} = \bigcup \mathcal{I}_s$ to be the range of the inputs. Moving forward, we assume that position encoding has been applied, allowing us to consider $\mathcal{X}^{(E)}$ as the input space. Consequently, we define the target space to be $\mathcal{C} = C(\mathcal{X}^{(E)}, \mathcal{Y})$, which denotes the space of continuous mappings between $\mathcal{X}^{(E)}$ and $\mathcal{Y}$.

### 4.1 Jackson-type approximation rate

To derive the Jackson-type approximation rate, it is necessary first to define the complexity measures to form an approximation space so that approximation rates can be obtained. We begin with the following representation theorem for the target space.

**Representation of the target space** $\mathcal{C}$   Given that the Transformer inherently captures both pairwise and pointwise relations among input tokens through attention and feed-forward components, respectively, we are motivated to establish the following representation theorem for the target space $\mathcal{C}$. This theorem demonstrates that every target can be exactly expressed in terms of pairwise and pointwise relations.

**Theorem 4.1** (Representation of the target space). *Consider $d$-dimensional, length $\tau$ input space $\mathcal{X}^{(E)}$ with position encoding added. Then, for any $\boldsymbol{H} \in C(\mathcal{X}^{(E)}, \mathcal{Y})$, there exists continuous functions $F \in C([0,1]^n, \mathbb{R})$, $f \in C(\mathcal{I}, [0,1]^n)$ and $\rho \in C(\mathcal{I} \times \mathcal{I}, \mathbb{R})$ such that for all $t \in [\tau]$ we have*

$$H_t(\boldsymbol{x}) = F\left(\sum_{s=1}^{\tau} \sigma[\rho(x(t), x(\cdot))](s) f(x(s))\right), \tag{8}$$

*where $n = 2\tau d + 1$ and $\sigma$ is the softmax function. The proof is presented in Appendix A.2.*

We refer to $\rho$ as the temporal coupling component and $F$ and $f$ as the element-wise components. It's important to note that the functions $F$, $f$, and $\rho$ may not be uniquely determined. In Appendix B.1, we explore certain invariant properties associated with Equation 8. Additionally, we provide illustrative examples in Appendix B.1 where the target explicitly conforms to Equation 8. Leveraging this representation theorem, we define the following complexity measures for targets $\mathcal{H} \in \mathcal{C}$.

**Temporal Coupling Component**  Now, we discuss the complexity measures associated with the temporal coupling term $\rho(u, v)$ that is central to understanding the attention mechanism in the Transformer. We employ the proper orthogonal decomposition (POD) [3] to decompose the temporal coupling of $\rho$. This approach can be viewed as an extension of matrix singular value decomposition (SVD) to functions of two variables. We have the following decomposition: $\rho(u, v) = \sum_{k=1}^{\infty} \sigma_k \phi_k(u) \psi_k(v)$, where $\phi_k, \psi_k$ are orthonormal bases for $L^2(\mathcal{I})$, and the singular values $\sigma_k \geq 0$ are arranged in descending order. The bases $\phi_k$ and $\psi_k$ are of optimal choices, ensuring that $\hat{\rho}(u, v)$ satisfies:

$$\inf_{\text{rank}(\hat{\rho}) \leq r} \|\rho(u, v) - \hat{\rho}(u, v)\|_2^2 = \sum_{k=r+1}^{\infty} \sigma_k^2, \tag{9}$$

analogous to the Eckart-Young theorem for matrices, with the rank defined as the number of terms in the POD decomposition. This implies that the approximation quality of $\rho(u, v)$ can be measured by the decay rate of its singular values. This motivates our following definition of complexity measure regarding $\rho$. Let $\alpha > 1/2$ be a constant, and $\{\sigma_i^{(\rho)}\}$ be singular values of $\rho$ under POD. We define the complexity measure of $\boldsymbol{H}$ by

$$C_1^{(\alpha)}(\boldsymbol{H}) = \inf_{F, f, \rho} \inf \left\{ c : \sigma_s^{(\rho)} \leq c s^{-\alpha}, s \geq 1 \right\}, \tag{10}$$

where the first infimum is taken over all $F, f, \rho$ such that Equation (8) holds. In particular, $C_1^{(g)}(\boldsymbol{H}) < \infty$ if it has a representation in the form (8) with $\rho$ having fast decaying singular values.

**Element-wise Component**  Now, we introduce the complexity measure for approximating the element-wise components $F$ and $f$. Let $\mathcal{F}^{(m_{\text{FF}})}$ be a hypothesis space comprising feed-forward neural networks with a budget of $m_{\text{FF}}$, representing parameters such as width or depth. We assume the existence of $\beta > 0$ such that:

$$\inf_{\hat{f} \in \mathcal{F}^{(m_{\text{FF}})}} \left\| f - \hat{f} \right\| \leq \frac{C_{\text{FF}}^{(\beta)}(f)}{m_{\text{FF}}^{\beta}}. \tag{11}$$

Here, $C_{\text{FF}}$ represents a complexity measure of $f$ corresponding to its approximation by $\mathcal{F}^{(m_{\text{FF}})}$. It is essential to emphasize that we are assuming the existence of pre-existing approximation rate results for the feed-forward component. For instance, in Barron [2], $\mathcal{F}^{(m_{\text{FF}})}$ is considered to be one-layer neural networks with sigmoidal activation, where $m_{FF}$ corresponds to the width of the network. By adopting this result, we have $\beta = 1/2$ and $C_{\text{FF}}^{(\beta)}(f)$ is a moment of the Fourier transform of $f$. For shallow ReLU networks commonly used in Transformers, Klusowski & Barron [18] demonstrate $\beta = 1/2 + 1/n$, where $n$ is the input dimension of the neural network. We maintain the generality of Equation (11). This enables us to substitute any other relevant estimates from the mentioned references. This flexibility allows for a broader application of the complexity measure in different scenarios and settings. Next, we proceed to define the complexity measure for $\boldsymbol{H} \in \mathcal{H}$. This measure considers all the components that need to be approximated using the feed-forward network.

$$C_2^{(\beta)}(\boldsymbol{H}, k) = \inf_{F, f, \rho} \left( C_{\text{FF}}^{(\beta)}(F) + C_{\text{FF}}^{(\beta)}(f) + C_{\text{FF}}^{(\beta)}(\rho, k) \right), \tag{12}$$

where $C_{\text{FF}}^{(\beta)}(\rho, k) = \sum_{i=1}^{k}(C_{\text{FF}}^{(\beta)}(\phi_i) + C_{\text{FF}}^{(\beta)}(\psi_i))$ considers the approximation of the approximation of the POD bases $\phi_i$ and $\psi_i$ for the target function $\rho$. Notably, this complexity measure is dependent on a parameter $k$, which determines the number of bases we want to consider in the approximation of $\rho$. Finally, we define a complexity measure considering the norm of the target.

$$C_0(\boldsymbol{H}) = \inf_{F,f,\rho} \left\{ K_F \|f\|_{\infty} (\sup_i \|\psi_i\|_{\infty} + \|\phi_i\|_{\infty}), 1 \right\}, \tag{13}$$

where $K_F$ is the Lipschitz constant of $F$ and $\phi_i, \psi_i$ are POD bases of $\rho$.

**Approximation Rates**  Combining the complexity measures Equations (10) and (13) and Equation (12) discussed above, we define the approximation spaces, which consists of targets that have finite complexity measures:

$$\mathcal{C}^{(\alpha,\beta)} = \{\boldsymbol{H} \in \mathcal{C} : C_0(\boldsymbol{H}) + C_1^{(\alpha)}(\boldsymbol{H}) + C_2^{(\beta)}(\boldsymbol{H}, k) < \infty, \ k \geq 1\}. \tag{14}$$

One can understand this as an analog of the classical Sobolev spaces for polynomial approximation but adapted to the Transformer hypothesis space. Note that $\mathcal{C}^{(\alpha,\beta)}$ is also dense in general continuous target space $C(\mathcal{X}^{(E)}, \mathcal{Y})$ when $n$ sufficiently large (See Appendix A.2). We are now ready to present the main result of this paper.

**Theorem 4.2** (Jackson-type approximation rates for the Transformer). *Consider sequences with a fixed length $\tau$. Suppose the target $\boldsymbol{H} \in \mathcal{C}^{(\alpha,\beta)}$ has a representation in the form Equation* (8) *with $F \in C([0,1]^{n'}, \mathbb{R})$ and $f \in C(\mathcal{I}, [0,1]^{n'})$. Let the hidden dimension of the Transformer be $n = 2 * m_h + m_v$, with $m_v \geq n'$. Then, we have:*

$$\inf_{\hat{\boldsymbol{H}} \in \mathcal{H}^{(m)}} \int_{\mathcal{I}} \sum_t^{\tau} \left| H_t(\boldsymbol{x}) - \hat{H}_t(\boldsymbol{x}) \right| d\boldsymbol{x} \leq \tau^2 C_0(\boldsymbol{H}) \left( \frac{C_1^{(\alpha)}(\boldsymbol{H})}{m_h^{2\alpha-1}} + \frac{C_2^{(\beta)}(\boldsymbol{H}, m_h)}{m_{\text{FF}}^{\beta}} \cdot (m_h)^{\beta+1} \right),$$

*where $m = (n, m_h, m_v, m_{\text{FF}})$ is the approximation budget, and $C_0, C_1^{(\alpha)} C_2^{(\beta)}$ are complexity measures of $\boldsymbol{H}$ defined in* (13), (10) *and* (12), *respectively. The proof is presented in Appendix A.3.*

Here, $m_h$ denotes the hidden dimension of the attention mechanism, essentially the size of the query and key vectors, while $m_{\text{FF}}$ represents the complexity measure of the pointwise feed-forward network employed in the Transformer. We first consider how the attention mechanism $\hat{\rho}(x(t), x(s))$ approximates the temporal coupling term $\rho$. By setting $[W_Q]_k = e_k$ and $[W_K]_k = e_{k+m_h+1}$, we can write $\hat{\rho}$ into $\hat{\rho}(x(t), x(s)) = (W_Q \hat{f}(x(t)))^\top W_K \hat{f}(x(s)) = \sum_{k=1}^{m_h} \hat{\phi}_k(x(t))\hat{\psi}_k(x(s))$, where $\hat{\phi}_k, \hat{\psi}_k : \mathcal{I} \to \mathbb{R}$ for $k \in [m_h]$ are components of $\hat{f}$ and are represented by the feed-forward network. This suggests that the approximation of $\rho$ by $\hat{\rho}$ is a low-rank approximation as discussed in Equation (9). When we increase $m_h$, the first term in the error rate that considers the POD decomposition decreases since there are more basis functions included. However, in scenarios where $m_{\text{FF}}$ remains unchanged, the second term in the error bound will increase. It is important to highlight that this error increment pertains only to the error bound, not to the best approximation error, which does not necessarily become worse when increasing the approximation budget. When $m_h$ increases, there are more basis functions that need to be approximated by the feed-forward components; thus, the error bound converges when both $m_h \to \infty$ and $m_{\text{FF}}^{\beta}/m_h^{\beta+1} \to \infty$. In Appendix B.2, we provide a synthetic example to illustrate the above discussion.

The complexity measure $C_2(\cdot)$ accounts for the quality of approximation using feed-forward networks. On the other hand, $C_1(\cdot)$ is the most interesting part, which concerns the internal structure of the attention mechanism. It tells us that if a target can be written in form (8) with $\rho(u, v)$, then it can be efficiently approximated with small $m_h$ if $\rho(u, v)$ has fast decaying singular values. This decay condition can be understood as effectively a low-rank condition on $(u, v) \mapsto \rho(u, v)$, analogous to the familiar concept for low-rank approximations of matrices. These observations provide important insights into the structure, bias, and limitations of the Transformer.

## 5  Numerical Demonstrations

In this section, we present numerical examples to demonstrate our approximation rate results. We begin with synthetic examples, where we can specify the singular value decay patterns, thus validating

the Jackson-type approximation rates in Theorem 4.2. Then, we turn to a practical example involving a Vision Transformer (ViT) model applied to the CIFAR10 dataset, where we do not have direct access to the singular values. We will discuss methods to estimate the singular value decaying pattern.

## 5.1 Practical Example

We next analyze a practical example, focusing on the Vision Transformer (ViT) model with the CIFAR10 dataset. In this scenario, we do have direct access to the temporal coupling term $\rho$ of the target relationship. However, we demonstrate that as we train a sequence of models with increasing attention dimension $m_h$, the singular values of $\hat{\rho}$ converges, implying the decaying pattern of $\rho$. Our first step is to estimate the rank of $\hat{\rho}$ from sampled data. Subsequently, we examine the singular value decay pattern, and the error changes when altering the attention head size $m_h$.

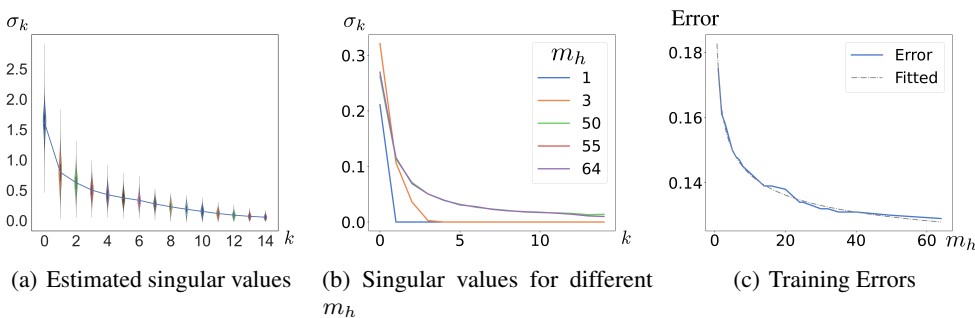

(a) Estimated singular values     (b) Singular values for different $m_h$     (c) Training Errors

Figure 1: (a) is the estimated singular value of the attention matrix over a set of inputs for $m_h = 64$. The violin plot shows the distribution of each singular value. (b) plots the estimated singular values for models with different $m_h$. (c) plots the training error against $m_h$.

**Estimate the Rank of $\hat{\rho}$**   Given a trained model, it is hard to directly compute the rank of $\hat{\rho}(u, v)$. Instead, we examine the attention matrix $\hat{\boldsymbol{\rho}}(\boldsymbol{x}) \in \mathbb{R}^{\tau \times \tau}$ where $[\hat{\boldsymbol{\rho}}(\boldsymbol{x})]_{t,s} = \rho(x(t), x(s))$. This is essentially a sample from $\hat{\rho}$. By analyzing the rank of these sampled matrices, we can estimate the rank of $\hat{\rho}$. According to Braun [5], the singular values of the matrix $\hat{\boldsymbol{\rho}}(\boldsymbol{x})$ exhibit the same decay pattern as those of $\hat{\rho}(u, v)$. Consequently, we can approximate the singular values of $\hat{\rho}(u, v)$ by averaging the singular values of $\hat{\boldsymbol{\rho}}(\boldsymbol{x})$ across various inputs. In Figure 1(a), we numerically estimate the singular values of $\hat{\rho}$ in the ViT-B_16 model [11]. We observe that the singular values tend to be more concentrated, suggesting that we can effectively estimate the rank of $\hat{\rho}(u, v)$ by evaluating it at sampled inputs.

We next estimate the singular value decaying pattern of the temporal coupling term $\rho$ in the target. In Figure 1(b), we analyze the singular values of the $\hat{\rho}$ for ViT models with varying values of $m_h$. We estimate the singular values by averaging over a set of inputs. We observe that as $m_h$ increases and reaches a sufficiently large value, the decaying pattern of the singular values starts to converge. As an example, Figure 1(b) plots the estimated singular values for the first head of the last layer for models with different $m_h$. This convergence suggests that the rank of the attention matrix becomes representative of the actual rank of $\rho$ in the target. Consequently, it suggests the presence of a low-rank structure in real-world datasets. In Figure 1(c), we plot the training error as an estimation of the approximation error. The plot reveals that the error decreases as $m_h$ increases, following a power law decaying pattern $O\left(1/m_h^{0.27}\right)$. This indicates that the target indeed exhibits a low-rank structure, and the pattern of error decay aligns with our approximation rate presented in Theorem 4.2. This illustrates that while our theorem is formulated based on a simplified scenario, the phenomenon of low rank is also observable in real-world datasets and models.

## 6 Comparison with RNN

Based on the approximation results in Theorem 4.2, this section presents a comparison between the Transformer and RNN. Our comparison centers on how each model is affected by the alterations in the temporal structures of sequential relationships. We primarily investigate two distinct temporal

structures: temporal ordering and temporal mixing, as they have varying impacts on the performance of each architecture. Our approach involves manipulating the temporal structures within the sequential relationships and evaluating how each architecture is affected by these changes. Proof are presented in Appendix D.2 and Appendix D.3.

## 6.1 Temporal Ordering Structure

We first analyze how the Transformer and RNN handle the change in temporal ordering of the sequential relationship. Empirically, it is observed that in certain contexts, the ordering of inputs does not significantly impact the relationships. For instance, in NLP applications, altering the word order in a sentence often does not drastically change its meaning. Similarly, in the ViT model, the arrangement of image patches typically does not substantially affect the outcome. However, in specific applications such as time series analysis, temporal ordering plays a crucial role, as the target relationships are governed by the ordering of the sequence. To alter the temporal order of target $\boldsymbol{H}$, we apply a fixed permutation $p$ and define the new target as $\tilde{H}_t(\boldsymbol{x} \circ p) = H_t(\boldsymbol{x})$. This permutes the input but keeps the output unchanged, resulting in a change to the temporal ordering.

We start by considering the RNN, which is affected by the change in temporal ordering. As demonstrated in Li et al. [22], a linear RNN is represented by the form $\hat{H}_t(\boldsymbol{x}) = \sum_s c^\top e^{Ws} U x(t-s)$. When we employ it to approximate linear targets represented by $H_t(\boldsymbol{x}) = \sum_s \rho(s) x(t-s)$, the complexity measures of the RNN $C_{\mathrm{RNN}}(\boldsymbol{H})$ is determined by both decay speed and magnitude of $\rho(s)$. We use $\mathcal{C}_{\mathrm{RNN}}$ to denote the approximation space for the RNN. (See Appendix D.1). We next show that the RNN is affected by the change in temporal ordering.

**Proposition 6.1.** *Let $\boldsymbol{H} \in \mathcal{C}_{RNN}$ and $p$ be a fixed permutation, such that there exists $t'$ with $p(t') > t'$. Suppose $\tilde{\boldsymbol{H}}$ is defined by $\tilde{H}_t(\boldsymbol{x} \circ p) = H_t(\boldsymbol{x})$. Then $\tilde{\boldsymbol{H}} \notin \mathcal{C}_{RNN}$.*

This proposition shows that the altered target $\tilde{\boldsymbol{H}}$ no longer belongs to the approximation space for RNN. This lies in the fact that RNN can only handle causal targets, where $y(t)$ does not depend on future inputs. However, the permuted target $\tilde{\boldsymbol{x}}$ is no longer causal, making the RNN incapable of learning such relationships. We next show that the Transformer, in contrast, remains unaffected by changes in temporal ordering.

**Proposition 6.2.** *Let $\boldsymbol{H} \in \mathcal{C}^{(\alpha,\beta)}$ and $p$ be a fixed permutation. Suppose $\tilde{\boldsymbol{H}}$ is defined by $\tilde{H}_t(\boldsymbol{x} \circ p) = H_t(\boldsymbol{x})$. Then $\tilde{\boldsymbol{H}}$ have same complexity measures with $\boldsymbol{H}$ for the complexity measures $C_0, C_1, C_2$ defined in Equation* (13),(10) *and* (12).

This proposition shows that the altered target $\tilde{\boldsymbol{H}}$ maintains the same complexity measures as the original target. This observation implies that the Transformer's approximation capability is not affected by alterations in temporal ordering. This point is further substantiated by our testing of the Transformer on real-world datasets, as illustrated in Table 1. We consider the ViT model on the CIFAR10 dataset and the base Transformer structure [30] on the WMT2014 English-German dataset. To alter the temporal ordering of the target relationship, we fix a permutation of indices denoted as $p$ and apply it to all inputs while keeping the output unchanged. The experimental results provide evidence that the performance of the Transformer is unaffected by the temporal ordering of the target relationships.

|          | CIFAR10 (*Acc.*) | ENG-DE (*BLUE*) |
|----------|------------------|-----------------|
| Original | 0.98             | 26.85           |
| Altered  | 0.96             | 25.91           |

Table 1: Numerical results of the Transformer on original and altered targets. The altered target is constructed by permuting the entire input dataset while keeping the output unchanged.

## 6.2 Temporal Mixing Structure

In this section, we explore how mixing elements from different time indices can affect the performance of the Transformer and RNN. Temporal mixing refers to the idea of blending information from various time indices, often through operations like convolution, which can alter the temporal structure. To

illustrate, consider a linear relationship represented as $H_t(\boldsymbol{x}) = \sum_s \rho(s)x(t-s)$. Now, imagine we apply a weighted sum of the input sequence $\boldsymbol{x}$ using a filter $\theta$ to get an altered input. We denote this operation as $\tilde{\boldsymbol{x}} = \theta * \boldsymbol{x}$, where $(\theta * \boldsymbol{x})[t] = \sum_{s=0}^{l-1} \theta(s)\boldsymbol{x}(t+s)$. This mixes the information in the sequence from different time indices. In this case, we define the altered target as $\tilde{H}_t(\boldsymbol{x}) = H_t(\theta * \boldsymbol{x}) = \sum_s \tilde{\rho}(s)x(t-s)$, where $\tilde{\rho} = \theta * \rho$ is the altered kernel (See Appendix D.3). This scenario often arises in signal processing and data analysis. For example, when dealing with a noisy input signal $\boldsymbol{x}$, one capproach is to apply a moving average filter to smooth it out. This filtering process involves mixing information from different time indices, which can significantly affect the behavior of target relationships. In the following sections, we will explore how such temporal mixing affects the temporal structures in sequential relationships. We begin by examining the linear RNN.

**Proposition 6.3.** *Let $\boldsymbol{H} \in \mathcal{C}_{RNN}$ associated with representation $\rho$, such that $|\rho(s)| \leq e^{-\frac{s}{\gamma}}$ for some $\gamma > 0$. Let $\theta$ be a length $l$ filter such that $\|\theta\|_1 \leq 1$. Suppose $\tilde{\boldsymbol{H}}$ is defined by $\tilde{H}_t(\boldsymbol{x}) = H_t(\theta * \boldsymbol{x})$. Then we have both $C_{RNN}(\boldsymbol{H}) \leq \gamma$ and $C_{RNN}(\tilde{\boldsymbol{H}}) \leq \gamma$, where $C_{RNN}$ is the complexity measure of the RNN (See Equation (75)).*

This proposition shows that under temporal mixing $\theta$ with certain conditions, the complexity measure of the altered target $\tilde{\boldsymbol{H}}$ does not worsen. As a result, the performance of the RNN is unaffected in such cases. However, performance of the Transformer can be impacted by temporal mixing in the target relationship. Consider $\boldsymbol{H} \in \mathcal{C}^{(\alpha,\beta)}$ and an altered target $\tilde{H}_t(\boldsymbol{x}) = H_t(\theta * \boldsymbol{x})$. This alteration can affect the complexity measures for $\tilde{F}, \tilde{f}$ and $\tilde{\rho}$ in $\tilde{\boldsymbol{H}}$. In Appendix D.3, we present an example where the rank of $\tilde{\rho}$ increases compared to $\rho$, leading to a performance drop in the Transformer. Numerical results in Table 1 also indicate that temporal mixing influences the Transformer's approximation capability. Preprocessing the input to mitigate this temporal mixing could potentially enhance performance, we leave this as a future direction. The following numerical examples demonstrate the above discussions. We conduct numerical experiments to substantiate the discussions above. These experiments focus on a linear target relationship defined as $H_t(\boldsymbol{x}) = \sum_s e^{-s}x(t-s)$. To manipulate the temporal ordering, we apply permute the function $e^{-s}$. Additionally, for introducing temporal mixing, the input is convolved with a randomly generated filter. Both the RNN and the Transformer are employed to learn these targets. Detailed experimental settings are discussed in the Appendix C. The results are presented in Table 2. The bold font indicates a performance drop in the architecture under the corresponding modification of temporal structures. It is observed that the performance of the Transformer remains unaffected by changes in the temporal ordering structure; however, it is impacted by temporal mixing. In contrast, the RNN exhibits opposite behaviors. This highlights that neither architecture consistently outperforms the other, as they each adapt to different types of temporal structures.

|  | Temporal Ordering | | Temporal Mixing | |
| --- | --- | --- | --- | --- |
|  | **RNN** | Trans. | RNN | **Trans.** |
| Original | **$1.02e{-}7$** | $2.18e{-}5$ | $1.02e{-}7$ | **$2.18e{-}5$** |
| Altered | **$3.57e{-}2$** | $2.51e{-}5$ | $1.58e{-}7$ | **$2.39e{-}4$** |

Table 2: The table presents the MSE values for both RNN and the Transformer under different alterations of temporal structures.

# 7 Conclusion

In this paper, we have developed Jackson-type approximation rates for the Transformer in a simplified setting. An important outcome of our work is the identification of complexity measures and approximation space defined in Equation (14). This, in turn, enables us to derive explicit Jackson-type approximation rate results in Theorem 4.2. Our rate results suggest that the Transformer performs well when the temporal coupling of the target exhibits a low-rank pattern. The experiments presented in Section 5 showcase the existence of low-rank patterns in real-world applications. Furthermore, the comparisons with RNNs underscore the specific temporal structures that each model handles efficiently. Future research directions involve extending the analysis to multi-headed attention and deeper Transformers. Additionally, we aim to investigate the potential benefits of removing temporal mixing in the input to enhance the performance of the Transformer.

## Acknowledgement

This research is part of the programme DesCartes and is supported by the National Research Foundation, Prime Minister's Office, Singapore under its Campus for Research Excellence and Technological Enterprise (CREATE) programme. QL acknowledges support by the National Research Foundation, Singapore, under the NRF fellowship (project No. NRF-NRFF13-2021-0005).

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

# A  Proof of Theorems

## A.1  Permutation Equivariance of the Hypothesis Space $\mathcal{H}$

Let $p : [\tau] \to [\tau]$ be a permutation. Let's consider an $\hat{H} \in \mathcal{H}$ with permuted inputs:

$$\hat{H}_t(\boldsymbol{x} \circ p) = \hat{F}\left( \sum_{s=1}^{\tau} \sigma[(W_Q \hat{f}(x(p(t))))^\top W_K \hat{f}(\cdot)](p(s)) \cdot W_V \hat{f}(x(p(s))) \right), \tag{15}$$

Since the permutation of index $s$ does not affect the sum, we have

$$= \hat{F}\left( \sum_{s=1}^{\tau} \sigma[(W_Q \hat{f}(x(p(t))))^\top W_K \hat{f}(\cdot)](s) \cdot W_V \hat{f}(x(s)) \right), \tag{16}$$

$$= \hat{H}_{p(t)}(\boldsymbol{x}). \tag{17}$$

This shows that the Transformer hypothesis $\mathcal{H}$ is permutation equivariant.

## A.2  Density of the Target Space $\mathcal{C}$

In this section, we show that the target space $\mathcal{C}$ defined in Equation (8) is dense in the general continuous target space $C(\mathcal{X}^{(E)}, \mathcal{Y})$. This ensures that the space defined is general enough to approximate arbitrary continuous targets.

To begin with, we first introduce the following representation theorem for multivariate functions.

**Theorem A.1** (Kolmogorov Representation Theorem). *[26] Let $\mathcal{I}_1, \ldots, \mathcal{I}_\tau$ be compact $d$ dimensional metric spaces. Then there are continuous functions $\psi_{qs} : \mathcal{I}_s \to [0, 1]$ and continuous function $g_q : [0, \tau] \to \mathbb{R}$, such that any continuous function $f : \prod \mathcal{I}_i \to \mathbb{R}$ can be represented as*

$$f(x_1, \ldots, x_\tau) = \sum_{q=0}^{2\tau d} g_q \left( \sum_{s=1}^{\tau} \psi_{qs}(x_s) \right). \tag{18}$$

This theorem states that any $\tau$ variable functions can be decomposed into superpositions of one variable function.

Now, we discuss how the pointwise functions on a sequence can apply different mappings on different time indices.

**Proposition A.2.** *Suppose we have $\tau$ variables in disjoint domains, where $x_1 \in \mathcal{I}_1, \cdot, x_\tau \in \mathcal{I}_\tau$ and $\mathcal{I}_1, \ldots, \mathcal{I}_\tau$ are all disjoint. We consider the following vector-valued function*

$$f(x_1, \ldots, x_\tau) = (f_1(x_1), \ldots, f_\tau(x_\tau)), \tag{19}$$

*where $f_s : \mathcal{I}_s \to \mathbb{R}$ for $s \in [\tau]$. We can indeed define a pointwise function $g : \bigcup \mathcal{I}_s \to \mathbb{R}$ to represent $f$. Since all $\mathcal{I}_s$ are disjoint, we can define a piecewise function $g$, where $g(x_s) = f_s(x_s)$ holds for all $s \in [\tau]$.*

This proposition demonstrates that for a sequence input $\boldsymbol{x}$ with appropriate positional encodings, where $x(i)$ and $x(j)$ belong to disjoint sets, a pointwise function can represent distinct mappings for these elements. We are now ready to present the following theorem.

**Theorem A.3.** *Consider $d$-dimensional, length $\tau$ input space $\mathcal{X}^{(E)}$ with position encoding added. Then, for any $\boldsymbol{H} \in C(\mathcal{X}^{(E)}, \mathcal{Y})$, there exists continuous functions $F \in C([0, 1]^n, \mathbb{R})$, $f \in C(\mathcal{I}, [0, 1]^n)$ and $\rho \in C(\mathcal{I} \times \mathcal{I}, \mathbb{R})$ such that for all $t \in [\tau]$ we have*

$$H_t(\boldsymbol{x}) = F\left( \sum_{s=1}^{\tau} \sigma[\rho(x(t), x(\cdot))](s) f(x(s)) \right), \tag{20}$$

*where $n = 2\tau d + 1$ and $\sigma$ is the softmax function.*

*Proof.* Based on the representation for continuous function Theorem A.1, we can decompose $\boldsymbol{H}$ into

$$H_t(\boldsymbol{x}) = \sum_{q=0}^{2\tau d} g_q^{(t)} \left( \sum_{s=1}^{\tau} \psi_{q,s}(x_s) \right),\tag{21}$$

where $\psi_{q,s}^{(\tau)} : \mathcal{I}_s \to [0,1]$ and $g_q^{(t)} : [0,\tau] \to \mathbb{R}$ are continuous functions. We next construct $F$, $f$ and $\rho$ to make $\boldsymbol{H}$ satisfy this form. Firstly, since $\mathcal{I}_s$ are disjoint we can define proper piecewise function $\rho$ such that $\sigma[\rho(x(t),\cdot)](x(s)) = \begin{cases} \frac{2}{\tau+1} & t = s \\ \frac{1}{\tau+1} & t \neq s \end{cases}$, which simplifies Equation (8) to

$$H_t(\boldsymbol{x}) = F\left( \frac{1}{\tau+1}\Big( f(x(t)) + \sum_{s=1}^{\tau} f(x(s)) \Big) \right).\tag{22}$$

Next, based on Proposition A.2, we let the pointwise function $f : \mathcal{I} \to [0,1]^n$ to apply different mappings for each $x(s)$, such that

$$f : x(s) \mapsto \left( \hat{\psi}_{0,s}(x_s), \dots, \hat{\psi}_{2\tau d,s}(x_s), b_s \right),\tag{23}$$

where $b_s \in [0,1]$ are different constants. We then have that

$$f(x(t)) + \sum_{s=1}^{\tau} f(x(s)) = \left( \hat{\psi}_{0,t}(x_t) + \sum_{s=1}^{\tau} \hat{\psi}_{0,s}(x_s), \dots, \hat{\psi}_{2\tau d,t}(x_t) + \sum_{s=1}^{\tau} \hat{\psi}_{2\tau d,s}(x_s), b_t + \sum_{s=1}^{\tau} b_s \right).\tag{24}$$

Here, $b_t$ performs as a shifting to make the range of Equation (24) disjoint for different $t$. Again, based on Proposition A.2, we can define $F$ to have individual mappings for different $t$. We first define $F_1 : [0,\tau+1]^n \to [0,\tau]^n$ to be

$$F_1 : u(t) \mapsto \left( u_1(t) - \hat{\psi}_{0,t}(x_t), \dots, u_{2\tau d+1}(t) - \hat{\psi}_{2\tau d,t}(x_t), u_n(t) - \sum_{s=1}^{\tau} b_s \right),\tag{25}$$

where $u_i$ denotes the $i$-th dimension of $u$. Next, define $F_2 : [0,\tau]^n \to \mathbb{R}$ to be

$$F_2 : u(t) \mapsto \sum_{q=0}^{2\tau d} g_q^{(t)} \left( u_q(t) \right).\tag{26}$$

Finally, let $F(u) = F_2 \circ F_1((\tau+1)u)$. Substitute into Equation (22) we get that

$$H_t(\boldsymbol{x}) = \sum_{q=0}^{2\tau d} g_q^{(t)} \left( \sum_{s=1}^{\tau} \psi_{q,s}(x_s) \right).\tag{27}$$

$\square$

This theorem shows that the target space $\mathcal{C}$ is, in fact, a representation of the general continuous target space $C(\mathcal{X}^{(E)}, \mathcal{Y})$. In particular, since $\mathcal{I}$ is a compact metric space, the complexity measures of each component $g_q^{(s)}$ and $\psi_{q,s}$ used in the construction of $H$ have finite complexity measures $C_0, C_1$ and $C_2$. This implies that the approximation space $\mathcal{C}^{(\alpha,\beta)}$ is also dense.

### A.3 Proof of Jackson-type approximation rate Theorem 4.2

Now, we present the proof of the Jackson-type approximation rates.

**Lemma A.4.** *The following inequality will be used to prove the theorem.*

$$\left| ab - \hat{a}\hat{b} \right| \leq |a|\,|b - \hat{b}| + |\hat{b}||a - \hat{a}|\tag{28}$$

$$\leq |a|\,|b - \hat{b}| + |b|\,|a - \hat{a}| + |a - \hat{a}|\,|b - \hat{b}|\tag{29}$$

*Proof.* Proof of Theorem 4.2 Since there are various components in the model, we will consider each of them separately. Firstly let's consider the approximation of $F$.

Let

$$h_t(\boldsymbol{x}) = \sum_s^\tau \sigma(\rho(x(t), x(s))) f(x(s)) \tag{30}$$

and

$$\hat{h}_t(\boldsymbol{x}) = \sum_s^\tau \sigma(\hat{\rho}(x(t), x(s))) \hat{f}(x(s)) \tag{31}$$

we have

$$\left| F(h) - \hat{F}(\hat{h}) \right| = \left| F(h) - F(\hat{h}) + F(\hat{h}) - \hat{F}(\hat{h}) \right| \tag{32}$$

$$\leq K_F \left| h - \hat{h} \right| + \left| F(\hat{h}) - \hat{F}(\hat{h}) \right|. \tag{33}$$

Let's consider $\left| F(\hat{h}) - \hat{F}(\hat{h}) \right|$ first. Since $\mathcal{I}$ is a compact domain and $\hat{h}$ is continuous, its range $\hat{h}(\mathcal{I})$ is a compact set. For all $x \in \hat{h}(\mathcal{I})$ we have $\left| F(\hat{h}(\boldsymbol{x})) - \hat{F}(\hat{h}(\boldsymbol{x})) \right| \leq \| F - \hat{F} \|_{L^\infty(\hat{h}(\mathcal{I}))}$. This implies that

$$\sum_t^\tau \int_{\mathcal{I}} \left| F(\hat{h}_t(\boldsymbol{x})) - \hat{F}(\hat{h}_t(\boldsymbol{x})) \right| d\boldsymbol{x} \leq \sum_t^\tau \int_{\mathcal{I}} d\boldsymbol{x} \cdot \left\| F - \hat{F} \right\|_{L^\infty(\hat{h}(\mathcal{I}))} \tag{34}$$

$$= \tau \| \mathcal{I} \| \left\| F - \hat{F} \right\|_{L^\infty(\hat{h}(\mathcal{I}))} \tag{35}$$

where $|\mathcal{I}|$ denotes the volume of $\mathcal{I}$. Recall that $\hat{h}_t(\boldsymbol{x}) = \sum_s^\tau \sigma(\hat{\rho}(x(t), x(s))) \hat{\rho}(x(s))$, we may assume the parameters of $\hat{f}$ is bounded such that $\hat{f}(x(s)) \subset [0, 1]$. By the property of Softmax we have $\hat{h}(\mathcal{I}) \subset [0, 1]$, thus,

$$\leq \tau |\mathcal{I}| \left\| F - \hat{F} \right\|_{L^\infty([0,1])} \leq \tau \left\| F - \hat{F} \right\|. \tag{36}$$

Next, let's consider the first term of (32),

$$\left| h - \hat{h} \right| \leq \sum_s^\tau \left| \sigma(\rho(x_t, x_s)) f(x_s) - \sigma(\hat{\rho}(x_t, x_s)) \hat{f}(x_s) \right| \tag{37}$$

$$\leq \sum_s^\tau |f| \left| \sigma(\rho(x_t, x_s)) - \sigma(\hat{\rho}(x_t, x_s)) \right| + |\sigma(\hat{\rho}(x_t, x_s))| \left| f - \hat{f} \right| \tag{38}$$

Since Softmax is Lipschitz continuous and bounded by 1, we have that

$$\leq \sum_s^\tau |f| |\rho - \hat{\rho}| + \sum_s^\tau \left| f - \hat{f} \right|. \tag{39}$$

The second term is an approximation with neural networks

$$\int \sum_{t,s}^\tau \left| f(x_s) - \hat{f}(x_s) \right| d\boldsymbol{x} \leq \tau^2 \left\| f - \hat{f} \right\|. \tag{40}$$

Now we remain to derive a bound for $|\rho - \hat{\rho}|$. Let $\tilde{\rho}$ be a $m_h$ term truncation of POD expansion of $\rho$, then

$$|\rho - \hat{\rho}| = |\rho - \tilde{\rho} + \tilde{\rho} - \hat{\rho}| \tag{41}$$

$$\leq |\rho - \tilde{\rho}| + |\tilde{\rho} - \hat{\rho}|. \tag{42}$$

The first term is the error estimates using POD, which says that

$$\int \sum_{t,s}^{\tau} |\rho(x_t, x_s) - \tilde{\rho}(x_t, x_s)|^2 \, d\boldsymbol{x} = \tau^2 \, \|\rho - \hat{\rho}\|_2^2 \tag{43}$$

$$\leq \tau^2 \sum_{i=m_h+1}^{\infty} \sigma_i^2. \tag{44}$$

The second term is again approximation with neural works, which we have

$$\sum_{i=1}^{m_h} \left| \phi_i \psi_i - \hat{\phi}_i \hat{\psi}_i \right| \leq \sum_{i=1}^{m_h} |\phi_i| \, |\psi_i - \hat{\psi}_i| + |\psi_i| \left| \phi_i - \hat{\phi}_i \right| + \left| \phi_i - \hat{\phi}_i \right| \, |\psi_i - \hat{\psi}_i| \tag{45}$$

Thus, combining all the inequalities above, we have that

$$\int \sum_{t}^{\tau} \left| H_t(\boldsymbol{x}) - \hat{H}_t(\boldsymbol{x}) \right| d\boldsymbol{x} \leq \tag{46}$$

$$K_F \sup |f| \tau^2 \left( \sum_{i=m_h+1}^{\infty} \sigma_i^2 + \sum_{i=1}^{m_h} \sup |\phi_i| \left\| \psi_i - \hat{\psi}_i \right\| + \sup |\psi_i| \left\| \phi_i - \hat{\phi}_i \right\| + \left\| \phi_i - \hat{\phi}_i \right\| \left\| \psi_i - \hat{\psi}_i \right\| \right) \tag{47}$$

$$+ \tau^2 \left\| F - \hat{F} \right\| + \tau^2 \left\| f - \hat{f} \right\| \tag{48}$$

By substituting the complexity measure of $\boldsymbol{H}$ to R.H.S., we have that

$$R.H.S. \leq \tau^2 \mathcal{C}_0(\boldsymbol{H}) \left( \frac{C_1^{(\alpha)}(\boldsymbol{H})}{m_h^{2\alpha-1}} + \frac{C_2^{(\beta)}(\boldsymbol{H})}{m_{\text{FF}}^{\beta}} \cdot (m_h)^{\beta+1} \right). \tag{49}$$

$\square$

This completes the proof.

# B    Extra discussions

## B.1    Numerical Examples of the Target Form Equation (8)

In this section, we provide numerical examples that follow the target form Equation (8):

$$H_t(\boldsymbol{x}) = F \left( \sum_{s=1}^{\tau} \sigma[\rho(x(t), x(\cdot))](s) f(x(s)) \right).$$

This form, in fact, is not a unique representation; in other words, for $\boldsymbol{H}_1 = \boldsymbol{H}_2$ we may not have $F_1 = F_2, f_1 = f_2$ and $\rho_1 = \rho_2$. However, we have the following propositions that characterize the invariant properties of the target form.

**Proposition B.1.** *Suppose $\boldsymbol{H}_1$ and $\boldsymbol{H}_2$ are in the form of Equation* (8)*, then the following properties hold.*

1. *If $\boldsymbol{H}_1(\boldsymbol{x}) = \boldsymbol{H}_2(\boldsymbol{x})$ holds for all $\boldsymbol{x}$, then $F_1 \circ f_1 = F_2 \circ f_2$.*

2. *Suppose $\sigma$ is hard-max function. For non constant $\boldsymbol{H}_1, \boldsymbol{H}_2 \in \tilde{\mathcal{C}}$, if $\boldsymbol{H}_1(\boldsymbol{x}) = \boldsymbol{H}_2(\boldsymbol{x})$ holds for all $\boldsymbol{x}$, then $\underset{x(s)}{\arg\max}[\rho_1(x(t), x(s))] = \underset{x(s)}{\arg\max}[\rho_2(x(t), x(s))]$ if the argmax is unique and $F_1 \circ f_1, F_2 \circ f_2$ are injections.*

3. *Suppose $\sigma$ is softmax function. For non constant $\boldsymbol{H}_1, \boldsymbol{H}_2$, if $\boldsymbol{H}_1(\boldsymbol{x}) = \boldsymbol{H}_2(\boldsymbol{x})$ holds for all $\boldsymbol{x}$ and $\rho_1, \rho_2$ are unbounded, then $\underset{x(s)}{\arg\max}[\rho_1(x(t), x(s))] = \underset{x(s)}{\arg\max}[\rho_2(x(t), x(s))]$ if the argmax is unique and $F_1 \circ f_1, F_2 \circ f_2$ are injections.*

*Proof.* By considering a constant input sequence $\boldsymbol{x} = (x, \ldots, x)$, we get $F_1 \circ f_1(x) = F_2 \circ f_2(x)$ for all $x$.

Suppose $\arg\max[\rho_1(x(t), x(s))] = c_1$ and $\arg\max[\rho_2(x(t), x(s))] = c_2$, then $\boldsymbol{H}_1(\boldsymbol{x}) = \boldsymbol{H}_2(\boldsymbol{x})$ implies $F_1 \circ \rho_1(c_1) = F_2 \circ \rho_2(c_2)$. By the injection assumption, we have $c_1 = c_2$, which implies the argmax are equal.

When the normalization is softmax, since $\rho$ is unbounded, we consider a sequence of inputs $\{\boldsymbol{x}^{(i)}\}$ such that the $\rho(x^{(i)}(t), x^{(i)}(s))$ goes to infinity, which became the same as the previous hard-max case. $\qquad\square$

Although these properties need to satisfy certain conditions to be theoretically correct, we empirically found they generally hold true in real applications.

Next, we demonstrate these concepts through numerical examples by utilizing the Transformer model to learn different targets. We consider the following simplified version of a single layer Transformer

$$\hat{H}_t(\boldsymbol{x}) = \hat{F}\left(W_V \hat{f}(\boldsymbol{x}) \cdot \sigma[(W_Q \hat{f}(\boldsymbol{x}))^\top W_K \hat{f}(\boldsymbol{x})]\right). \tag{50}$$

Here, $\boldsymbol{x} \in \mathbb{R}^{d \times \tau}$ is the input, $\hat{F}, \hat{f}$ are two nonlinear mappings applied column-wise, and $\sigma$ is the softmax function applied column-wise to the matrix.

**Nearest point to a set** In this example, we consider two sets of points $U \subset R^d$ and $V \subset R^d$ as inputs. For each point $u_t \in U$, we aim to determine the nearest point from set $V$. More specifically, we define an input sequence $x \in \mathcal{X} \subset \mathbb{R}^{2d \times \tau}$ in the form $x(t) = (u_t, v_t) \in R^{2d}$, where $u_t$ and $v_t$ are two $d$-dimensional points belonging to point set $U$ and $V$, respectively. The output is defined as

$$y(t) = \arg\min_{v \in V} |u(t) - v|. \tag{51}$$

This can be rewritten as

$$y(t) = \sum_{s=1}^{\tau} \sigma_H[-|u(t) - v(\cdot)|](s)v(s), \tag{52}$$

$$\tag{53}$$

where $\sigma_H : \mathbb{R}^\tau \to \mathbb{R}^\tau$ is the hard-max function. Note this equation conforms with the form in Equation (8) with $F \circ f$ begin identity and $\rho(u, v) = |u - v|$. For the numerical example, we consider $U, V \subset \mathbb{R}^2$ and each have 6 points, and train a Transformer model to learn this target and examine the learned model.

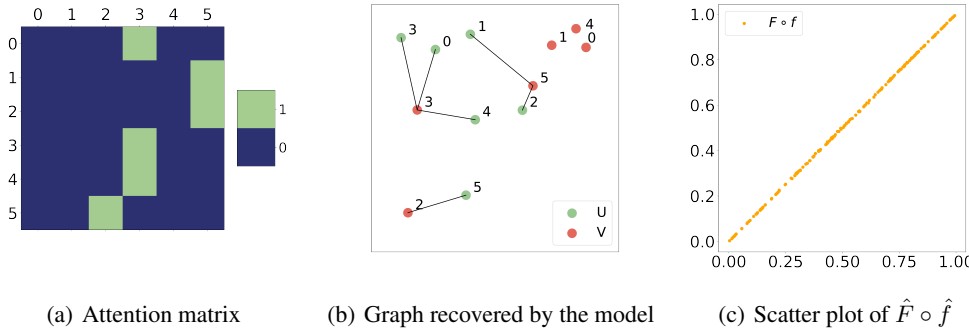

(a) Attention matrix      (b) Graph recovered by the model      (c) Scatter plot of $\hat{F} \circ \hat{f}$

Figure 2: For Figures (a) and (b) we examine a particular instance of the input $\boldsymbol{x}$. Figure (a) plots the attention matrix $A$, while Figure (b) illustrates the learned relationships, with green points and red points representing points from set $U$ and $V$, respectively. Figure (c) is the scatter plot of $F \circ f(\boldsymbol{x})$ for randomly generated inputs $\boldsymbol{x}$.

In Figure 2(a), we plot the attention matrix $A$ of a particular instance of the input $\boldsymbol{x}$. For Figure 2(b), we visualize the point set $U, V$ based on their coordinates and establish connections according

to the values in $A$. Specifically, two points $u_i$ and $v_j$ are connected based on the value of $A_{ij}$. This visualization reveals exactly that each green point in set $U$ is linked to the nearest point in $V$, demonstrating the model's ability to recover the target relationship accurately. Furthermore, for the target form Equation (52), $F \circ f$ in this case is an identity function. As shown in Figure 2(c), $\hat{F} \circ \hat{f}$ is also found to learn an identity function. This observation is in accordance with Proposition B.1, despite the theoretical assumptions that may not fully apply.

**Weighted average by weight** In this scenario, we are given sequences of point masses as inputs. Our objective is to compute the center of gravity for each sequence. These point masses are in $\mathbb{R}^d$ space, each with a mass $m \in \mathbb{R}$.

Consider a sequence containing $\tau$ points. We extend this to a sequence of $\tau + 1$ points by adding an extra point $x_{\text{pred}} \notin \mathcal{X}$ at the beginning of the sequence, serving as a prediction token. Consequently, the input sequence is represented as:

$$\boldsymbol{x} = \left(x_{\text{pred}}, \begin{pmatrix} x_1 \\ m_1 \end{pmatrix}, \ldots, \begin{pmatrix} x_\tau \\ m_\tau \end{pmatrix}\right). \tag{54}$$

The corresponding output sequence is formulated as:

$$\boldsymbol{y} = (|\bar{x}|, |x_1|, \ldots, |x_\tau|), \tag{55}$$

where $x_{\text{pred}}$ is mapped to $\bar{x}$, and the other points remain unchanged. $|\cdot|$ denotes the Euclidean norm. The center of gravity $\bar{x}$ is defined by the equation:

$$\bar{x} := \sum_{s=1}^{\tau} \frac{m_s}{\sum_{s'} m_{s'}} x_s. \tag{56}$$

We may consider several sequence segments of point masses, denoted as $\boldsymbol{x}_1$ to $\boldsymbol{x}_n$, which may vary in length. These sequences are concatenated to form an input sequence $\boldsymbol{x}$. Correspondingly, the output $y$ concatenates $\boldsymbol{y}_1$ to $\boldsymbol{y}_n$.

As an example, consider the following input-output pair:

$$\boldsymbol{x} = \left(x_{\text{pred}}, \begin{pmatrix} x_1 \\ m_1 \end{pmatrix}, \begin{pmatrix} x_2 \\ m_2 \end{pmatrix}, \begin{pmatrix} x_3 \\ m_3 \end{pmatrix}, x_{\text{pred}}, \begin{pmatrix} x_4 \\ m_4 \end{pmatrix}, \begin{pmatrix} x_5 \\ m_5 \end{pmatrix}\right). \tag{57}$$

$$\boldsymbol{y} = \left(\left|\frac{\sum_{s=1}^3 m_s x_s}{\sum_{s'=1}^3 m_{s'}}\right|, |x_1|, |x_2|, |x_3|, \left|\frac{\sum_{s=4}^5 m_s x_s}{\sum_{s'=4}^5 m_{s'}}\right|, |x_4|, |x_5|\right). \tag{58}$$

In this example, the output sequence $\boldsymbol{y}$ includes the norm of the computed centers of gravity for each concatenated segment of the input sequence $\boldsymbol{x}$, where each segment is separated by the prediction token $x_{\text{pred}}$. This can be formulated as target form Equation (8) such that

$$y(t) = \begin{cases} \left\|\sum_{s \in \mathcal{I}_x} \frac{m_s}{\sum_{s' \in \mathcal{I}_x} m_{s'}} x(s)\right\| & x(t) = x_{\text{pred}} \\ \left\|\sum_{s=1}^{\tau} \mathbf{1}_{s=t} \, x(s)\right\| & \text{otherwise} \end{cases}, \tag{59}$$

where $\mathcal{I}_x$ denotes the index of the sequence segment corresponding to $x_{\text{pred}}$. The weighted output here depends on the prediction token, such that it only takes the weighted average within the corresponding sequence. For numerical illustration, we consider input $\boldsymbol{x}$ of length 10, containing sequence segments of length at least 2. We apply the single-layer Transformer model to learn this target.

In Figure 3, we analyze an input $\boldsymbol{x}$ comprising two sequence segment, $\boldsymbol{x}_1$ and $\boldsymbol{x}_2$, with lengths of 3 and 5, respectively. In this scenario, $x(0)$ and $x(5)$ are prediction tokens. The attention matrix behaves as expected: the outputs $y(0)$ and $y(5)$ are influenced predominantly by the points within their respective sequences, and the values are approximately the normalized weight. Furthermore, since the remaining outputs are the same as the inputs, the attention matrix values for these elements are confined to the diagonal, which means that the output at that point is only determined by itself.

Furthermore, in Figure 3(c) we observe that $\hat{F} \circ \hat{f}$ can successfully recover $F \circ f$ in the target which is a norm of the $\mathbb{R}^2$ vector.

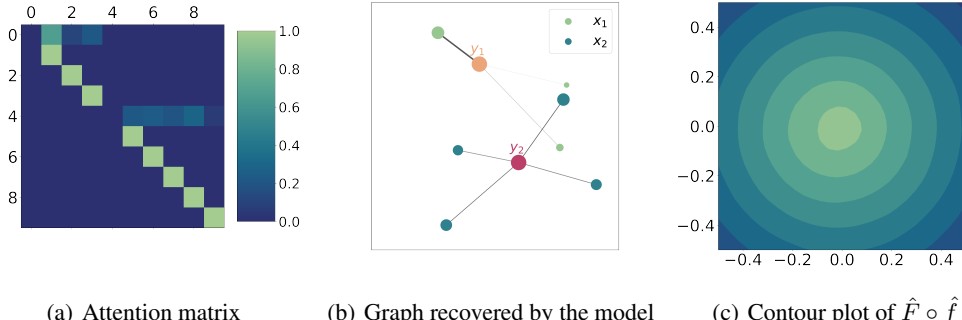

(a) Attention matrix      (b) Graph recovered by the model      (c) Contour plot of $\hat{F} \circ \hat{f}$

Figure 3: Figure (a) plots the attention matrix $A$, while Figure (b) is the illustration of the learned relationship. In this instance, there are two sequences, $x_1$ and $x_2$, each connected to their respective predictions. The color of the connecting lines represents the corresponding values in $A$. Figure (c) presents the contour plot of $F \circ f(x)$, generated for a set of random inputs $x$.

**Weighted average by selection** The previous calculated the weighted average by specifying the weight of each point. Now, we consider another way to calculate a weighted average where we specify a subset of points we are going to take the average. Suppose we have a length $\tau$ sequence of $\mathbb{R}^d$ vectors $v \in \mathbb{R}^{d \times \tau}$. We also have a sequence of vectors $s \in [\tau]^{k \times \tau}$ consisting of sequences of indices. We stack $v$ and $s$ to form the input sequence $x \in \mathbb{R}^{(d+k) \times \tau}$. This extends the qSA task proposed in Sanford et al. [27].

The output $y(t)$ is defined as

$$y(t) = \frac{1}{k} \sum_{s \in s(t)} v_s. \tag{60}$$

This can be written in the target form Equation (8) such that $F \circ f$ is identity. Depending on the frequency of $s$ appears in the index vector $s(t)$, we have that

$$\rho(x(t), x(s)) \in \left\{ 0, \frac{1}{k}, \frac{2}{k}, \dots, 1 \right\} \tag{61}$$

As an example, we have the following input-output pairs.

$$v = (v_1, v_2, v_3),$$

$$s = \left( \begin{pmatrix} 1 \\ 1 \\ 2 \end{pmatrix}, \begin{pmatrix} 1 \\ 2 \\ 3 \end{pmatrix}, \begin{pmatrix} 3 \\ 3 \\ 3 \end{pmatrix} \right). \tag{62}$$

$$y = \frac{1}{3} \Big( 2v_1 + v_2, \ v_1 + v_2 + v_3, \ 3v_3 \Big).$$

In our numerical experiments, we set $\tau = 8$, $d = 2$, and $k = 3$. This configuration implies that both the input and output sequences have a length of 8, and the output $y(t)$ is computed as the average of three inputs, as indices designated by $s(t)$. For this particular setting we have $\rho \in \{0, \frac{1}{2}, \frac{2}{3}, 1\}$. Again, we use a single-layer transformer model to learn this target.

In Figure 4 we examine a particular input, where

$$s = \left( \begin{pmatrix} 4 \\ 5 \\ 5 \end{pmatrix}, \begin{pmatrix} 4 \\ 4 \\ 4 \end{pmatrix}, \begin{pmatrix} 1 \\ 1 \\ 7 \end{pmatrix}, \begin{pmatrix} 5 \\ 5 \\ 5 \end{pmatrix}, \begin{pmatrix} 0 \\ 3 \\ 7 \end{pmatrix}, \begin{pmatrix} 0 \\ 6 \\ 7 \end{pmatrix}, \begin{pmatrix} 2 \\ 2 \\ 2 \end{pmatrix}, \begin{pmatrix} 2 \\ 3 \\ 4 \end{pmatrix} \right). \tag{63}$$

In Figure 4(a), we observe that the attention matrix accurately reconstructs $\rho$, as demonstrated in Equation (61). Figure 4(b) illustrates the graph representation of this input-output correlation based on the matrix $A$. In this graph, an output $i$ and an input $j$ are connected by the value of $A_{ij}$. The figure highlights the model's capability to recover the relationship: Output $y(1), y(3), y(6)$ each rely on a singular input value, output $y(0), y(2)$ are calculated as the average of two values, and output $y(4), y(5), y(7)$ each represent an average over three values.

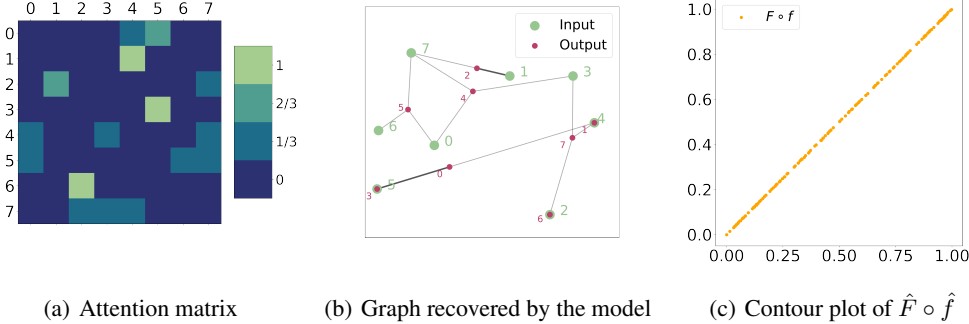

(a) Attention matrix     (b) Graph recovered by the model     (c) Contour plot of $\hat{F} \circ \hat{f}$

Figure 4: Figure (a) plots the attention matrix $A$, while Figure (b) illustrates the learned relationship. The points in different colors refer to the input and output, respectively. They are connected based on the value of $A$. Figure (c) presents the scatter plot of $F \circ f(\boldsymbol{x})$, generated for a set of random inputs $\boldsymbol{x}$.

In summary, these examples demonstrate tasks that follow the form presented in Equation (8). Moreover, the numerical experiments validates Proposition B.1, where $F \circ f$ is always the same, and the attention matrix recovers the desired weights.

## B.2 Synthetic Examples

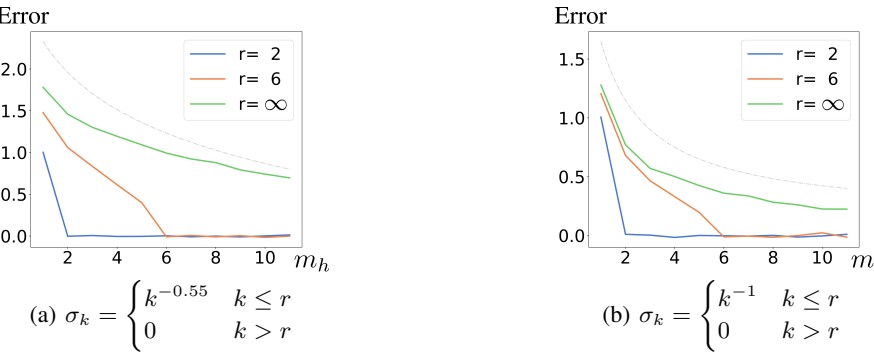

Figure 5: We consider two class of $\rho$ with different singular value decay rate $\alpha$ as indicated above. Here, $r$ denotes the rank of the target. For each (a) and (b), we consider three targets with $\rho$ having different ranks, where $r = 2, 6, \infty$. The figure plots the training error against $m_h$. Each colored line corresponds to a target with rank $r$ as indicated in the legend. The grey dotted line plots $m_h^{-(2\alpha-1)}$.

The Jackson-type approximation rate in Theorem 4.2 has the following prediction. For sequence relationships admitting a representation Equation (8), if its rank $r$ is finite, then perfect approximation is possible as long as $m_h \geq r$ with $m_{\mathrm{FF}}$ large enough. If $r = \infty$, then the approximation error is determined by decay of $\sigma_k$. Concretely, if $\sigma_k \sim k^{-\alpha}$, then the error decays like $m_h^{-(2\alpha-1)} + \text{constant}$, provided $m_{\mathrm{FF}}$ is sufficiently large. We numerically verify the prediction by constructing a set of targets using Equation (8). In this specific example, we set both $F$ and $f$ as the identity function. The temporal coupling term $\rho(u, v)$ is formulated using an orthonormal bases where $\phi_i, \psi_i \in \{\sqrt{2} \sin 2i\pi x : i \geq 1\}$, accompanied by singular values with various decaying patterns. Subsequently, we trained the Transformer as defined in Equation (5) to learn these targets. Figure 5 plots the training error against $m_h$, where (a) and (b) correspond to targets with different singular value decay patterns. We also consider different rank $r$ of the targets as plotted using lines with different colors. The error decay rates of (a) and (b) are different, which follows the decay rate determined by that of the singular values. Furthermore, for a fixed $m_h$, the error increases with the target's rank. The experiment results are consistent with our estimates in Theorem 4.2.

## C    Experiment settings

In this section, we summarize the settings for the numerical experiments.

**Experiments in Appendix B.2**    These experiments consider the synthetic targets with simplified Transformer models as defined in Equation (5). The target relationship is constructed using Equation (8):

$$H_t(\boldsymbol{x}) = F\left(\sum_{s=1}^{\tau} \sigma[\rho(x(t), x(\cdot))](s) f(x(s))\right). \tag{64}$$

We consider scalar inputs where $d = 1$ and $F(x) = f(x) = x$ are identity functions. $\rho(u, v)$ is constructed using the POD decomposition:

$$\rho(u, v) = \sum_{i=1}^{r} \sigma_i \phi_i(u) \psi_i(v). \tag{65}$$

We consider orthonormal bases $\phi_i, \psi_i \in \{\sqrt{2}\sin 2i\pi x : i \geq 1\}$. By doing this, we can specify $\sigma_i$, which are then exactly the singular values. The input sequence of length 16 is generated using a uniform distribution on $[0, 1]$. We use Transformers as defined in Equation (5) to learn these targets. The feed-forward part is constructed using a dense network with a width of 128 and a depth of 3 to ensure it has enough expressiveness. Moreover, we have $n = d_v = 32$ and $m_h$, which range from 1 to 16, to construct a model with different ranks. We use PyTorch default initialization and use normal training procedures with Adam Optimizer. We train enough epochs to ensure the loss does not decrease so that we can use the training error to estimate the approximation error.

**Experiments in Section 5.1**    In this example, we tested the ViT model on the CIFAR10 dataset. We consider the baseline model ViT_B16 [11]. This model configuration utilizes 12 heads, and each head is of dimension 64. In Figure 1(a), we plot the singular values distribution of the attention matrix (before Softmax normalization) from the first attention head. We then vary the size of each attention head from 1 to 64 while keeping other configurations unchanged. We follow the training procedure as described in [11] to train the modified models.

**Experiments in Section 6**

**Comparison of the Transformer and RNNs**    We consider the following linear target relationship:

$$H_t(\boldsymbol{x}) = \sum_{t=0}^{t} \rho(s) x(t - s), \tag{66}$$

with kernel $\rho(s) = \exp(-s)$. The input sequence of length 32 is generated using a uniform distribution on $[0, 1]$. For the RNN, we utilize a one-layer vanilla RNN architecture with 128 hidden units, employing a linear activation function. For the Transformer, we use the simplified structure as presented in Equation (8) with position encoding added. For parameter settings, we have $n = d_v = m_h = 32$. The feed-forward part is constructed using a dense network with a width 128 and a depth 3.

To change the temporal ordering, we permute the input while keeping the output unchanged, this leads to a change in the temporal ordering of the target relationship. For all the input sequences, we exchange the first 10 values with the last 10 values, resulting in a change in the underlying temporal structure. For the temporal mixing operation, we use a randomly generated length 5 filter and the operation $\divideontimes$ described in Section 6.2.

In Table 2, it is also interesting to note that for this target, the RNN performs better than the Transformer. This is potentially because temporal ordering is inherently important in this target, and the Transformer is not good at capturing such relationships. Some variants of the Transformer, such as Informer [36] and Autoformer [32], have been proposed specifically to address temporal relationships, particularly in sequence forecasting tasks. However, empirical results presented in Zeng et al. [35] indicate that these variants still cannot effectively capture temporal relationships.

As a comparison, we consider a different type of linear relationship where the temporal ordering is unimportant. We randomly generate the filter such that $\rho(s) \sim \mathcal{U}_{[0,1]}$. This does not have temporal

ordering since if we permute $\rho$, the distribution of it does not change. The results are shown in Table 3. We can observe that it is hard for RNN to learn this target. But the Transformer can learn this target, and a permutation of the input does not affect its performance.

|  | RNN | Trans. |
|---|---|---|
| Original | 5.58 | $5.43e-4$ |
| Permuted | 5.24 | $5.36e-4$ |

Table 3: The table presents the MSE values for both RNN and the Transformer datasets without temporal ordering.

**Real-world datasets for the Transformer**   In Table 1, we test the performance of the Transformer on a permuted dataset considering real-world examples. For the ViT experiment, we use the ViT_B16 model. For the WMT2014 English-German dataset, we use the original Transformer model as proposed in [30]. For both experiments, when testing on the permuted dataset, we initialize the model with parameters that are trained on the original dataset. To generate the permuted dataset, we fix a permutation such that the first 10 elements are shifted to the end of the sequence. This operation is applied to all the input sequences. Note that the permutation of the ViT dataset is before the addition of position encoding and after the convolution embedding.

# D   Detailed Dissuasion of Section 6

In this section, we provide additional discussions related to Section 6. We first review the approximation results for the RNN, the we provide proofs for the propositions presented in Section 6.

## D.1   Introduce the approximation results for the RNN

We here review the results of approximation results of RNN presented in Li et al. [23]. Here, we consider input and output space defined by

$$\mathcal{X} = \left\{ \boldsymbol{x} : x(s) \in [0,1]^d, s \in \mathbb{N}_{\geq 0} \right\} \tag{67}$$

and

$$\mathcal{Y} = \left\{ \boldsymbol{y} : y(s) \in \mathbb{R}, s \in \mathbb{N}_{\geq 0} \right\}. \tag{68}$$

There are infinite sequences with time indices starting from zero.

We consider the following dynamics for the linear RNN:

$$h(t) = Wh(t-1) + Ux(t) \tag{69}$$

$$y(t) = c^{\top} h(t), \tag{70}$$

where $h \in \mathbb{R}^n$ with $h(0) = 0$ is the hidden state of the RNN. $W \in \mathbb{R}^{n \times n}$, $U \in \mathbb{R}^{d \times n}$ and $c^{\top} \in \mathbb{R}^{n \times 1}$ are parameters. The approximation budget of the model is $n$, which is the size of its hidden state. We can explicitly solve this dynamic where we get

$$\hat{H}_t(\boldsymbol{x}) = \sum_{s=0}^{t} c^{\top} W^s U x(t-s). \tag{71}$$

Here, we assume the eigenvalue of $W$ has a negative real part to ensure it is stable when $s$ increases.

For the target, we consider a linear relationship represented by

$$H_t(\boldsymbol{x}) = \sum_{s=0}^{t} \rho(s) x(t-s), \tag{72}$$

where $\|\rho(s)\| \leq \infty$ is a unique representation of $\boldsymbol{H}$. Thus, the approximation capability of the RNN is characterized by the properties of $\rho$.

Note that $\hat{\rho}(s) = c^\top W^s U$ in the RNN exhibits an exponential decaying pattern. Thus, the target must also have a similar decay pattern to achieve a good approximation. We have precisely defined the following complexity measures for the RNN.

$$C_1(\boldsymbol{H}) = 1 + \inf\{\beta > 0 : \lim_{t \to \infty} e^{\frac{t}{\beta}} |\rho(t)| = 0\}. \tag{73}$$

This measures the decaying speed of the RNN, we assume $\rho$ exhibits an exponential decay, with decaying speed measured by $C_1$. The next complexity considers the norm of $\rho$:

$$C_2(\boldsymbol{H}) = \sup\{e^{\frac{t}{C_1(\boldsymbol{H})}} |\rho(t)| : t \geq 0\}. \tag{74}$$

Combined together, we define the complexity measure for RNN as

$$C_{\text{RNN}} = C_1(\boldsymbol{H}) \cdot C_2(\boldsymbol{H}) \tag{75}$$

This considers the magnitude of $\rho$, $C_2$ is small if $\rho$ does not have a large value far from the origin. We define the RNN approximation space to be targets that have finite complexity measures

$$\mathcal{C}_{\text{RNN}} = \{\boldsymbol{H} \text{ satisfies Equation (72)} : C_{\text{RNN}} < \infty\}. \tag{76}$$

The approximation rate of the RNN follows $\dfrac{d \cdot C_{\text{RNN}}}{m}$.

## D.2 Discussion of Section 6.1

**Proof of Proposition 6.1** We now present the proof for Proposition 6.1

*Proof.* Let $p : \mathbb{N}_{\geq 0} \to \mathbb{N}_{\geq 0}$ be a permutation on the time indices. We consider an altered target

$$\tilde{H}_t(\boldsymbol{x} \circ p) = H_t(\boldsymbol{x}) \tag{77}$$

We then have

$$\tilde{H}_t(\boldsymbol{x} \circ p) = \sum_{s=0}^{t} \tilde{\rho}(s) x(p(t - s)) \tag{78}$$

Let's consider the most simple case where $p$ only permute two time indices such that $p(t_1) = t_2$ and $t_2 > t_1$. We consider the output at $t_1$

$$H_{t_1}(\boldsymbol{x}) = \sum_{s=0}^{t_1} \rho(s) x(t_1 - s) \tag{79}$$

$$= \sum_{s=1}^{t_1} \rho(s) x(t_1 - s) + \rho(0) x(t_1). \tag{80}$$

However, for $\tilde{H}_{t_1}(\boldsymbol{x})$ we have

$$\tilde{H}_{t_1}(\boldsymbol{x} \circ p) = \sum_{s=0}^{t_1} \tilde{\rho}(s) x(p(t_1 - s)) \tag{81}$$

$$= \sum_{s=1}^{t_1} \tilde{\rho}(s) x(t_1 - s) + \tilde{\rho}(0) x(t_2). \tag{82}$$

Note the in this case the output at $t_1$ depends on a future value of input $x(t_2)$, this means that $\tilde{\boldsymbol{H}}$ is no longer causal. Consequently, $\tilde{\boldsymbol{H}}$ cannot be written in the form of Equation (72), thus not belong to the RNN approximation space. $\square$

**Proof of Proposition 6.2**   We next consider the Proposition 6.2

*Proof.* For a fixed permutation $p : \tau \to \tau$, we consider $\tilde{H}_t(\boldsymbol{x} \circ p) = H_t(\boldsymbol{x})$. Suppose the altered target has the following form:

$$\tilde{H}_t(\boldsymbol{x} \circ p) = \tilde{F}\left(\sum_{s=1}^{\tau} \sigma[\tilde{\rho}(x(p(t)), x(\cdot))](p(s))\tilde{f}(p(x(s)))\right), \tag{83}$$

we next show that $\tilde{F}, \tilde{f}$ and $\tilde{\rho}$ has same complexity measures as $F, f$ and $\rho$. Firstly, we note that we can remove the permutation on index $s$ because the ordering of a summation does not affect the result.

$$\tilde{H}_t(\boldsymbol{x} \circ p) = \tilde{F}\left(\sum_{s=1}^{\tau} \sigma[\tilde{\rho}(x(p(t)), x(\cdot))](s)\tilde{f}(s))\right). \tag{84}$$

We now consider the POD expansion of $\tilde{\rho}$ where

$$\tilde{\rho}(x(p(t)), x(s)) = \sum_{i=1}^{r} \sigma_i \tilde{\phi}_i(x(p(t))) \, \tilde{\psi}_i(x(s)). \tag{85}$$

Recall that our input space is defined as Equation (7). Where $x_s \in \mathcal{I}_s$ such that $\mathcal{I}_i$ and $\mathcal{I}_j$ are closed disjoint sets. Thus, for a pointwise function $\phi$, we can write it into the following piecewise form

$$\phi(x) = \phi^{(t)}(x) \quad x \in \mathcal{I}_t. \tag{86}$$

Let $\tilde{\phi}_i(x) = \phi_i^{(p(t))}(x)$ for $x \in \mathcal{I}_t$. This is essentially the permutation of the disjoint pieces of $\tilde{\phi}_i$. Thus we have

$$\tilde{\phi}_i(x(p(t))) = \phi_i^{(t)}(x(t)). \tag{87}$$

Consequently, we have $\tilde{\rho}(x(p(t)), x(s)) = \rho(x(t), x(s))$. Finally let $\tilde{F} = F$ and $\tilde{f} = f$ we achieved $\tilde{H}_t(\boldsymbol{x} \circ p) = H_t(\boldsymbol{x})$. Since $\tilde{F}$ and $\tilde{f}$ remain unchanged, we only need to consider the complexity measure for $\rho$. We note that the rank of the POD is unaffected. We remain with $C_{\mathrm{FF}}(\tilde{\phi}_i)$. The function $\tilde{\phi}_i$ is a permutation of pieces of the piecewise function $\phi_i$. We next discuss the kind of approximation schemes that $C_{\mathrm{FF}}(\tilde{\phi}_i)$ is unchanged under this operation.

Since the set $\mathcal{I}$ is disconnect, we apply Tietze extension theorem to extend $\phi_i$ to its convex hull, denoted as $\Phi_i$, such that $\Phi_i(x) = \phi_i(x)$ for $x \in \mathcal{I}$ and $\sup\{|\Phi_i(x)|\} = \sup\{|\phi_i(x)|\}$.

We first consider the polynomial approximation as we introduced in Section 3. In this case, we have $C_{\mathrm{FF}}(\phi_i) = \max_{r=1\ldots\alpha} \|\phi_i\|_{\infty}$. We next show that $C_{\mathrm{FF}}(\tilde{\phi}_i) = C_{\mathrm{FF}}(\phi_i)$. Since $\mathcal{I}$ is closed and $\tilde{\phi}$ is continuous, we apply Tietze extension theorem to extend $\tilde{\phi}_i$ to $\mathbb{R}^d$ denoted as $\tilde{\Phi}_i$, such that $\tilde{\Phi}_i(x) = \tilde{\phi}_i(x)$ for $x \in \mathcal{I}$ and $\sup\{|\tilde{\Phi}_i(x)|\} = \sup\{|\tilde{\phi}_i(x)|\}$. This implies that $C_{\mathrm{FF}}(\tilde{\phi}_i) = C_{\mathrm{FF}}(\phi_i)$. Thus, in the case of polynomial approximation, the complexity measures remain unchanged.

Next, we consider the approximation using the ReLu network. We assume $\phi_i$ to be Hölder continuous such that

$$|f(x) - f(y)| \le C \|x - y\|^{\alpha}. \tag{88}$$

The Tietze extension theorem states we can extend it to $\Phi_i$ with the same $C$ and $\alpha$.

As shown in Shen et al. [28], the approximation capability of a ReLU network on a Hölder function depends on $C$ and $\alpha$. Where functions with small $C, \alpha$ can be approximated more effectively. Since $\tilde{\phi}_i$ is a permutation of disconnected pieces of $\phi_i$, its Hölder constant remains unchanged. Thus, its $\tilde{\Phi}_i$ has same Hölder constant as $\Phi_i$, which implies the complexity measure is the same.

$\square$

In the above proof, we considered two different approximation schemes where the complexity measures of $\tilde{H}$ remain unchanged.

### D.3 Discussion for Section 6.2

**Proof of Proposition 6.3**   We first discuss Proposition 6.3. Consider a filter $\theta$ which is a length $l$ filter with $\|\theta\|_1 \leq 1$. The altered target is defined by

$$\tilde{H}_t(\boldsymbol{x}) = H_t(\theta * \boldsymbol{x}) = \sum_{s=0}^{t} \rho[s](\theta * x)[t-s]. \tag{89}$$

Rearrange the summation, and we get

$$\tilde{H}_t(\boldsymbol{x}) = \sum_{s=0}^{t} \rho[s](\theta * x)[t-s] \tag{90}$$

$$= \sum_{s=0}^{t} \rho(s) \sum_{s'=0}^{l-1} \theta[s']x(t-s+s') \tag{91}$$

$$= \sum_{s=0}^{t} \sum_{s'=0}^{l-1} \theta[s']\rho(s)x(t-s+s') \tag{92}$$

$$= \sum_{s=0}^{t} \sum_{s'=0}^{l-1} \theta[s']\rho[s+s']x[t-s] \tag{93}$$

$$= \sum_{s=0}^{t} (\theta * \rho)[s]x[t-s] \tag{94}$$

$$= \sum_{s=0}^{t} \tilde{\rho}[s]x[t-s]. \tag{95}$$

Thus, we have the altered kernel denoted as $\tilde{\rho} = \theta * \rho$.

*Proof.*   Firstly, for $|\rho(s)| \leq exp(-\frac{s}{\beta})$, by the definition of $C_1$ and $C_2$ we have $C_1(\boldsymbol{H}) \leq \beta$ and $C_2(\boldsymbol{H}) \leq 1$.

By assumption we have $\rho$ exhibits exponential decay such that $|\rho(s)| \leq e^{-\beta s}$ for some $\beta$. We then have

$$|\tilde{\rho}(s)| = |(\theta * \rho)[t]| = \sum_{s'=0}^{l-1} |\theta(s')\rho(s+s')| \leq e^{-\beta s} \|\theta\|_1 = e^{-\beta s}, \tag{96}$$

which shows that $\tilde{\rho}[t]$ is also bounded by $e^{-\beta s}$. This implies that $C_1(\tilde{\boldsymbol{H}}) \leq \beta$ and $C_2(\tilde{\boldsymbol{H}}) \leq 1$.

$\square$

**Transformer Affected by Temporal Mixing**   Finally, we present an example where a temporal mixing in the input will affect the target's rank. For simplicity we consider the case where $d = 1$ and $\tau = 2$, and $x_1, x_2 \in [0, 1]$. Note that here, we do not assume $x_1$ and $x_2$ belong to disjoint intervals for simplicity. For target in the form of Equation (8), we consider a temporal coupling term $\rho$ defined as

$$\rho(x_1, x_2) = 1 + \sqrt{2}\sin(2\pi x_1) \cdot \sqrt{2}\sin(2\pi x_2). \tag{97}$$

In this case, $\rho(u, v)$ is of rank 2 with both singular values equal to 1. Consider a new input $\tilde{\boldsymbol{x}} = (x_1, x_2 + x_1)$ with temporal mixing. We then have

$$\rho(x_1, x_2 + x_1) = 1 + \sqrt{2}\sin(2\pi x_1) \cdot \sqrt{2}\sin(2\pi(x_1, x)). \tag{98}$$

We numerically estimate the singular values of $\tilde{\rho}$ to get $\sigma_1 = 1.265$, $\sigma_2 = 0.5$ and $\sigma_3 = 0.4$. The rank of $\tilde{\rho}$ increases, and there are also extra bases included. Here, we only consider the rank $\rho$ from the same representation, while we do not exclude the possibility that there exist other representations that may have another rank pattern. Indeed, analyzing the temporal mixing in the Transformer is non-trivial; we only provided an example where the rank increases. Conversely, a suitable deconvolution process can also decrease the complexity measure of the target. We leave it as a future direction for analysis.

