# OpenReview forum: "Approximation Rate of the Transformer Architecture for Sequence Modeling"
_NeurIPS.cc/2024/Conference — NeurIPS 2024 poster_

### Official Review · Reviewer_fL7j · 2024-07-11

**Soundness:** 2
**Presentation:** 2
**Contribution:** 2
**Rating:** 5
**Confidence:** 3

**Summary:**

This study investigates a Jackson-type approximation rate for single-layer Transformers with one head, and compares their ability with RNNs, another nonlinear sequence-to-sequence map.

**Strengths:**

- The literature overview is concise
- The Jackson-type approximation rate for the Transformer is derived for the first time

**Weaknesses:**

The main theorem (Theorem 4.2) sounds trivial because the bound is a combination of the definitions of complexities $C^\alpha$ (Sobolev smoothness) and $C^\beta$ (Barron bound). The universality of Eq.8 (Theorem A.3) may sound non-trivial, but it is obtained by rewriting the Kolmogorov representation of continuous function (from Theorem A.1), which is much a stronger (but only existential) result.

It could be non-trivial if the authors could provide a similar bound in a constructive manner without using the magic argument of Kolmogorov.

**Questions:**

- Can the author provide a similar bound in a constructive manner without using the Kolmogorov theorem?
- Given a dataset, how to estimate $\alpha$ and $\beta$?

---

> ### Author Rebuttal · Authors · 2024-08-06
>
> 1. The main theorem (Theorem 4.2) sounds trivial because the bound is a combination of the definitions of complexities $C^\alpha$ (Sobolev smoothness) and $C^\beta$ (Barron bound).
>
>     - Firstly, we need to clarify that $ C^\alpha $ is not a Sobolev smoothness term. It is defined as the POD rank of the temporal coupling term $ \rho $ (Line 181 - Line 195). To the best of our knowledge, our approximation rate results (Theorem 4.2) have no similar results previously.
>
>     - Theorem 4.2 should not be dismissed as trivial. For Theorem 4.2, our analysis sheds light on the distinct roles of the feed-forward and attention components. The feed-forward component aims to approximate both the pointwise functions $F$ and $f$, as well as the POD bases of the temporal coupling component $\rho$. We substantiated the validity of our findings numerically in Section 5, demonstrating the existence of low-rank structures in Transformer approximations.
>
>     - Additionally, Theorem 4.2 provides insights into the differences between Transformers and traditional sequence modeling architectures like RNNs, discussed in Section 6. The complexity measures defined Section 4 are unaffected by permutation but is influenced by temporal mixing. This contrast implies the fundamental distinction of Transformers and RNNs in handling temporal relationships. In Section 6.1 we show that the RNN is proficient in handling temporal relationships with strong temporal ordering structure. While for relationship with minimal temporal ordering, the Transformers works better. Section 6.2 explores how temporal mixing can impact Transformer performance, whereas RNNs remain unaffected.
>
>     - Defining appropriate complexity measures and approximation spaces is crucial in approximation theory, offering insights into hypothesis spaces. A well-chosen approximation space can illuminate the capabilities of the hypothesis space. Referring to recent studies [1] and [2], which consider targets of the form $H_t(x) = \sum \rho(s)x(t-s)$, various complexity measures are defined for $\rho$ considering the architectures. In RNNs [1], these measures account for the smoothness and decay rate of $\rho$, whereas in CNNs [2], they focus on the sparsity of $\rho$. Different architectures reveal distinct approximation capabilities: RNNs excel with smooth and fast-decaying targets, while CNNs are effective with sparse targets. Our results also provides insights of the Transformer architectures, which are discussed in Section 5 and Section 6.
>
>
> [1] Li, Zhong, et al. "Approximation and optimization theory for linear continuous-time recurrent neural networks." Journal of Machine Learning Research 23.42 (2022): 1-85.
>
> [2] Jiang, Haotian, Zhong Li, and Qianxiao Li. "Approximation theory of convolutional architectures for time series modelling." International Conference on Machine Learning. PMLR, 2021.
>
>
>
> 2. Given a dataset, how to estimate $\alpha$ and $\beta$ ?
>
>     - Estimating $\alpha$ and $\beta$ directly from a dataset based on their definitions can be challenging. However, empirical estimation is feasible by training models of varying sizes on the dataset. In Section 5.1, we discuss how to estimate $\alpha$ in detail. The main idea is train models with varying $m_h$ and fit the error curve with $\frac{1}{m_h^\alpha} + c$ for an estimation of $\alpha$. It is important to highlight that insights derived from approximation results hold greater significance than exact values of approximation bounds. Specifically, in Section 6, we discuss the strengths and limitations of Transformers compared to RNNs, leveraging these approximation insights for discussion.
>
>
> 3. Can the author provide a similar bound in a constructive manner without using the Kolmogorov theorem?
>
>     - The approximation bound in Theorem 4.2 is independent of the Kolmogorov theorem. The Kolmogorov theorem is only used in Theorem 4.1 to establish that form (8) is general enough to represent any continuous target. The approximation bound in Theorem 4.2 considers targets of the form (8) where $F$, $f$, and $\rho$ are assumed to be general continuous function with finite complexity measures and not restricted by the Kolmogorov representation. Consequently, the results in Theorem 4.2 are not depend on the Kolmogorov theorem.

---

> ### Comment · Reviewer_fL7j · 2024-08-12
>
> Thank you for your clarifications. I will keep my score as is.
>
> > The approximation bound in Theorem 4.2 is independent of the Kolmogorov theorem.
>
> If so, it's misleading that Theorem 4.1 is put inside the Section 4.2, and it's even better if the paper is written completely without Kolmogorov.

---

> > ### Author Response · Authors · 2024-08-13
> >
> > Thanks for your response. We would like to clarify again that the purpose of Theorem 4.1 is to ensure that the target space considered in form (8) is large enough to represent any continuous sequence-to-sequence functions. We want to confirm that this target space is not restrictive. Then, in Theorem 4.2, we consider target functions of the form in (8) with regularities to develop the approximation rates. The Kolmogorov method is a proof technique for Theorem 4.1, there may also be other methods to prove the theorem, but this should not affect the results and logic flow of the paper.

---

### Official Review · Reviewer_GFZQ · 2024-07-13

**Soundness:** 3
**Presentation:** 2
**Contribution:** 3
**Rating:** 6
**Confidence:** 4

**Summary:**

This paper introduces a novel concept of complexity measures to construct approximation spaces for single-layer Transformers with one attention head, providing Jackson-type approximation rate results for target spaces that possess a representation theorem.

**Strengths:**

- The results in this paper are presented within a general framework using rigorous and elegant mathematical tools, offering a solid theoretical foundation for researchers interested in approximation.

- Their hypothesis of singular value decay pattern regarding the target space can be validated through the experiments detailed in Section 5. Furthermore, the hypothesis underscores the crucial role of pairwise coupling and low-rank structure.

**Weaknesses:**

The results presented are limited to 2-layer single-head Transformers, which restricts their applicability and insights into more common models such as multi-layer multi-head Transformers.

**Questions:**

- How can the results be generalized to analyze multi-layer multi-head Transformers? Will such generalization provide new insights or understanding?

- Although the discussion on the parameters' dependence on rank is provided, the factor $\tau^2$ in the RHS of the inequality in Theorem 4.2 appears suboptimal for approximating long sequences.

- It seems that much of the relevant literature on the approximation power of Transformers has been omitted. For example, [1][2][3].

[1] Giannou et al (2023). Looped Transformers as Programmable Computers.

[2] Bai et al (2023). Transformers as statisticians: Provable in-context learning with in-context algorithm selection.

[3] Wang \& E (2024). Understanding the Expressive Power and Mechanisms of Transformer for Sequence Modeling.

**Limitations:**

See Weaknesses.

---

> ### Author Rebuttal · Authors · 2024-08-06
>
> 1. How can the results be generalized to analyze multi-layer multi-head Transformers? Will such generalization provide new insights or understanding?
>
>     - Firstly, our rate still applies to multi-layer and multi-head Transformers, serving as an upper bound. This is because our single-layer, single-head architecture represents the simplest form of multi-layer, multi-head Transformers. However, increasing the number of layers and heads can potentially yield more refined bounds. Theoretical analysis of the relationship with depth and the number of heads requires consideration of a more intricate approximation space, which includes defining suitable target form with complexity measures that account for depth and the multi-head structure.
>
>     - While our theoretical results are based on simplified architectures, empirical verification in Section 5 and Section 6 confirms that these insights (be specific) hold for multi-layer architectures as well. In Section 5, we demonstrated the existence of low-rank structures in multi-layer Transformer architectures. Section 6 discuss both the strengths and limitations of Transformers compared to RNNs. We verified our statements regarding general multi-layer and multi-head structures.
>
> 2. Although the discussion on the parameters' dependence on rank is provided, the factor $\tau^2$ in the RHS of the inequality in Theorem 4.2 appears suboptimal for approximating long sequences.
>
>     - The quadratic scaling of Transformers in sequence length is a well-known issue. The factor of $\tau^2$ arises because the size of attention matrix scales as $O(\tau^2)$ with the sequence length,leading to quadratic scaling in computation time. This scaling also affects approximations, as described by Equation (43), where the approximation of the attention matrix involves both temporal directions $t$ and $s$, resulting in approximation errors that scale with $\tau^2$.
>
> 3. It seems that much of the relevant literature on the approximation power of Transformers has been omitted. For example, [1][2][3].
>
>    - [1] considers a special setting regarding expressiveness, demonstrating that Transformers can represent any computer program. [2] and [3] explore target relationships with certain special structures. Thank you for highlighting these references. We will include them in the related work section.
>
> [1] Giannou et al (2023). Looped Transformers as Programmable Computers.
>
> [2] Bai et al (2023). Transformers as statisticians: Provable in-context learning with in-context algorithm selection.
>
> [3] Wang & E (2024). Understanding the Expressive Power and Mechanisms of Transformer for Sequence Modeling.

---

> > ### Comment · Reviewer_GFZQ · 2024-08-12
> >
> > Thank you for your clarifications. I will maintain my score.

---

### Official Review · Reviewer_YNqb · 2024-07-17

**Soundness:** 3
**Presentation:** 3
**Contribution:** 3
**Rating:** 8
**Confidence:** 4

**Summary:**

The study explores the theoretical aspects of Transformer architectures in sequence modeling, particularly focusing on approximation rates for sequence-to-sequence relationships. A representation theorem is established, introducing novel complexity measures that analyze interactions among input tokens, culminating in a Jackson-type approximation rate estimate for Transformers.

**Strengths:**

This study enhances the understanding of Transformer's approximation rate and gives concrete comparisons with traditional models like recurrent neural networks.

**Weaknesses:**

The paper deviates from the standard Transformer architecture by requiring a neural network layer before the attention mechanism to implement the Kolmogorov Representation Theorem, potentially inheriting the theorem's limitations.

**Questions:**

Q1. In the context of POD, it is my understanding that having rho with fast decaying singular values allows for the complexity measure of H to be constant. However, this paper employs a specific construction to derive equation (22) for sigma(rho) as mentioned in line 492.
Could we still say that rho has fast decaying singular values in this case? Please provide a specific explanation.

Q2. It's unclear how to construct the F1 function that realizes equation (25). Could you explain in detail?

Q3. I cannot  understand the flow from equation (22) to (25).
Eequation (24) (f(x_t)+sum_s f(x_s)) is constructed using equation (22), which is realized by using a specific attention, and then f(x_t) is removed in equation (25).
Why not just construct an attention, average pooling, that constitutes sum_s f(x_s) in Eq. (22)?

Q4. Theorem A.3 states that n=tau * (2 * tau * d+1)+1.
However, looking at equation (23), it appears that the output of f is only (2 * tau * d+1)-dimensional.
Where does n=tau * (2 * tau * d+1)+1 need to be?

Q.5 Considering the theoretical framework presented here which inherits limitations from the Kolmogorov representation theorem, could you specify what limitations might arise in the class of functions approximated by the model?
It is worth noting that identifying these limitations does not detract from the contributions of this paper.

**Limitations:**

The paper deviates from the standard Transformer architecture by requiring a neural network layer before the attention mechanism to implement the Kolmogorov Representation Theorem, potentially inheriting the theorem's limitations.

---

> ### Author Rebuttal · Authors · 2024-08-06
>
> 1. The paper deviates from the standard Transformer architecture by requiring a neural network layer before the attention mechanism to implement the Kolmogorov Representation Theorem, potentially inheriting the theorem's limitations.
>
>    - For our proposed architecture (5), the term $\hat h = \hat f \circ x$ is fed into the attention component, where $\hat f$ is a feed-forward network. This assumption is not particularly restrictive. Our architecture (5) can be viewed as a specific "slice" of the standard Transformer architecture. The standard Transformer follows a pattern of "Atten->FFN->Atten->FFN->...". Our formulation can be seen as focusing on the "FFN->Atten->FFN" part within this structure.
>
>
> 2.  In the context of POD, it is my understanding that having rho with fast decaying singular values allows for the complexity measure of H to be constant. However, this paper employs a specific construction to derive equation (22) for sigma(rho) as mentioned in line 492. Could we still say that rho has fast decaying singular values in this case? Please provide a specific explanation.
>
>    - Line 492 pertains to the proof of Theorem 4.1, demonstrating the equivalence of Equation (8) with the continuous function space $\mathcal C=C(\mathcal X^{(E)}, \mathcal Y)$. The proof is constructive, involving a specific construction of $\rho$ at Line 492. It is essential to emphasize that this construction is designed specifically for the proof of Theorem 4.1. When assuming a target in the form of Equation (8), we do not assume $\rho$ to have any specific form.
>
>    - For Theorem 4.2, we do not assume $\rho$ is fixed like the construction presented in Line 492. In Theorem 4.2, the target $H$ adopts the structure of Equation (8), where $\rho \in C(\mathcal I \times \mathcal I, \mathbb R)$ is a general continuous function that have finite complexity $C_1^{\alpha}$. Since $\rho$ is not restricted to a specific form, different target $H$ may correspond to varying patterns of singular value decay and thus different complexity measures.
>
>
>
> 3. It's unclear how to construct the F1 function that realizes equation (25). Could you explain in detail?
>
>    Firstly, for clarity, Equation (24) should be defined as $u(t) = f(x(t)) + \sum_{s=1}^\tau f(x(s))$. Next, it is observed that each $u(t)$ resides within disjoint cubes since $b_t$ are assumed to be distinct. Consequently, $F_1$ can be defined separately on each disjoint cube. For each $t$, the expression $F_1(u(t))$ is governed by Equation (25). Moreover, $u(t)$ is $n$-dimensional, where $u_i(t)$ denotes its $i$-th component.
>
>
> 4. I cannot understand the flow from equation (22) to (25). Eequation (24) (f(x_t)+sum_s f(x_s)) is constructed using equation (22), which is realized by using a specific attention, and then f(x_t) is removed in equation (25). Why not just construct an attention, average pooling, that constitutes sum_s f(x_s) in Eq. (22)?
>
>    - To clarify, Equation (24) should be defined as $u(t) = f(x(t)) + \sum_{s=1}^\tau f(x(s))$. Equation (25) does not remove $f(x_t)$;
>    rather, it is an intermediate definition of $F$ in Line 500.
>    - The proof aims to align the functions $F$, $f$, and $\rho$ in Equation (8) with Equation (18). Specifically, $\rho$ is defined at Line 492, and $f$ is given by Equation (23). The function $F$ is constructed as $F(u) = F_2 \circ F_1((\tau+1)u)$, where its definition is decomposed into two distinct functions for clarity. By substituting these expressions for $f$ and $F$ into Equation (22), the proof achieves an exact correspondence with Equation (18).
>
>
>
> 5. Theorem A.3 states that n=tau * (2 * tau * d+1)+1. However, looking at equation (23), it appears that the output of f is only (2 * tau * d+1)-dimensional. Where does n=tau * (2 * tau * d+1)+1 need to be?
>    - Thanks for pointing out. This is indeed a typo, the correct formula should be $n=(2 * \tau * d+1)$.
>
>
> 6.  Considering the theoretical framework presented here which inherits limitations from the Kolmogorov representation theorem, could you specify what limitations might arise in the class of functions approximated by the model? It is worth noting that identifying these limitations does not detract from the contributions of this paper.
>
>    - A techniqual limitation of the function space arises because a given $H$ may correspond to different sets of $F$, $f$, and $\rho$, implying non-uniqueness of the form (8). This aspect is reflected in the definitions of complexity measures in (10), (12), and (13), where we take the infimum over all possible $F$, $f$, and $\rho$.
>
>    - In Section 6, we also discuss limitations arising from target form (8). Proposition 6.2 states that permutation of input sequences does not alter the complexity measures, suggesting that temporal ordering is insignificant within this function space. This implies that Transformers excel in handling sequential relationships with minimal temporal dependencies, as detailed in Section 6.1. However, the complexity of the function space can be significantly influenced by temporal mixing. As discussed in Section 6.2, this indicates that the performance of Transformers can be adversely affected by straightforward temporal mixing manipulations. To summary, these intrinsic structural limitations also implies approximation limitations of the Transformers.

---

> > ### Comment · Reviewer_YNqb · 2024-08-14
> >
> > Thank you for the response.
> > I will raise my score.

---

### Decision · Program_Chairs · 2024-09-25

**Decision:**

Accept (poster)

**Comment:**

The paper received favorable reviews. Reviewers acknowledge the novelty of the Jackson-type approximation rate for the Transformer and agree on the merits of this paper being presented at NeurIPS.